# Multimodal mechanisms of human centriole engagement and disengagement

Kei K Ito[1], Kasuga Takumi[1], Kyohei Matsuhashi[1], Hirokazu Sakamoto[2,3], Kaho Nagai[1], Masamitsu Fukuyama[1], Shohei Yamamoto[1], Takumi Chinen[1], Shoji Hata [1,3✉] & Daiju Kitagawa [1✉]

## Abstract

**Centrioles are unique cellular structures that replicate to produce identical copies, ensuring accurate chromosome segregation during mitosis. A new centriole, the "daughter", is assembled adjacent to an existing "mother" centriole. Only after the daughter centriole is fully developed as a complete replica, does it disengage and become the core of a new functional centrosome. The mechanisms preventing precocious disengagement of the immature daughter centriole have remained unclear. Here, we identify three key mechanisms maintaining mother–daughter centriole engagement: the cartwheel, the torus, and the pericentriolar material (PCM). Among these, the torus critically establishes the characteristic orthogonal engagement. We also demonstrate that engagement mediated by the cartwheel and torus is progressively released during centriole maturation. This release involves structural changes in the daughter, known as centriole blooming and distancing, respectively. Disrupting these structural transitions blocks subsequent steps, preventing centriole disengagement and centrosome conversion in the G1 phase. This study provides a comprehensive understanding of how the maturing daughter centriole progressively disengages from its mother through multiple steps, ensuring its complete structure and conversion into an independent centrosome.**

**Keywords** Centriole; Centrosome; Centriole Engagement; Centriole Disengagement; Centriole Distancing
**Subject Categories** Cell Adhesion, Polarity & Cytoskeleton; Cell Cycle

## Introduction

The centrosome is a dynamic organelle that serves as the primary microtubule-organizing center (MTOC) in animal cells (Nigg and Holland, 2018). Typically, a cell contains two centrosomes throughout the cell cycle. During mitosis, these centrosomes migrate to opposite sides of the cell, forming the mitotic spindle poles to ensure accurate chromosome segregation (Conduit et al, 2015). The presence of more than two centrosomes can lead to the formation of multipolar spindles, resulting in unequal chromosome segregation and potential aneuploidy (Ganem et al, 2009). To maintain proper centrosome number, centrosome duplication is tightly regulated and occurs only once per cell cycle (Nigg and Holland, 2018). Abnormal centrosome numbers are associated with diseases such as cancer and microcephaly (Nigg and Raff, 2009).

The regulation of centrosome number is primarily achieved through controlling its core component, the centriole (Nigg and Holland, 2018). In the G1 phase, cells contain two centrosomes, each with a mature centriole (mother centriole) surrounded by a protein matrix that includes the torus and the pericentriolar material (PCM). The torus, located just outside the centriole wall, supports centriole duplication, while the PCM surrounds the torus and organizes microtubules, facilitating MTOC activity (Pimenta-Marques and Bettencourt-Dias, 2020). In the early S phase, centriole duplication begins as the mother centriole assembles a cartwheel structure on its lateral surface, which serves as a scaffold for the formation of the daughter centriole wall (Gönczy and Hatzopoulos, 2019). The daughter centriole undergoes gradual maturation, first increasing in width (bloom phase) and then in length (elongation phase) (Laporte et al, 2024). The elongation continues until it reaches the same length as the mother centriole in mitosis (Kong et al, 2020). In the subsequent G1 phase, the mature daughter centriole disengages from its mother and converts into a functional centrosome with its own torus and PCM, completing centriole-to-centrosome conversion (Wang et al, 2011; Fu and Glover, 2016).

Precocious disengagement of a daughter centriole during mitosis results in its early conversion into a centrosome, increasing the number of centrosomes and potentially leading to abnormal spindle formation (Watanabe et al, 2019; Kim et al, 2019). If the daughter centriole disengages from its mother centriole before mitosis, it undergoes both early conversion and centriole reduplication. This combination leads to a higher number of centrosomes, resulting in more frequent mitotic errors (Ito et al, 2021). Several key proteins have been identified as crucial for maintaining centriole engagement. During mitosis, Cep57, which localizes on the mother centriole wall, maintains centriole engagement by stabilizing PCNT, a scaffold protein of the PCM (Watanabe et al, 2019; Kim et al, 2019). After

[1]Department of Physiological Chemistry, Graduate School of Pharmaceutical Science, The University of Tokyo, Bunkyo, Tokyo 113-0033, Japan. [2]Department of Pharmacology, Graduate School of Medicine, The University of Tokyo, Bunkyo, Tokyo 113-0033, Japan. [3]Precursory Research for Embryonic Science and Technology (PRESTO) Program, Japan Science and Technology Agency, Honcho Kawaguchi, 102-8666 Saitama, Japan. ✉E-mail: s.hata@mol.f.u-tokyo.ac.jp; dkitagawa@mol.f.u-tokyo.ac.jp

chromosome segregation, cleavage of PLK1-phosphorylated PCNT by the cysteine protease Separase triggers centriole disengagement (Kim et al, 2015; Matsuo et al, 2012; Lee and Rhee, 2012; Agircan and Schiebel, 2014). Meanwhile, during interphase, Cep57 and its paralog Cep57L1 redundantly maintain centriole engagement (Ito et al, 2021). Although the daughter centriole undergoes gradual structural changes while attached to the mother, the exact alterations in mother–daughter engagement from duplication to disengagement are not fully understood.

In this study, we uncover the stepwise changes in the configuration and mechanism of centriole engagement that allow the daughter centriole to disengage from its mother at G1 phase. Centriole engagement is initially maintained by the cartwheel structure immediately after daughter centriole formation in early S phase. This engagement is then reinforced by the torus and the PCM of the mother centriole, with the torus playing a crucial role in establishing the characteristic orthogonal engagement between the mother and the daughter centrioles. These engagement mechanisms are sequentially released in coordination with the structural changes of the daughter centriole during its maturation process. Disrupting the stepwise structural maturation of the daughter centriole at any stage prevents the associated transition of engagement mechanisms, thereby inhibiting the eventual disengagement and conversion of the daughter centriole into a centrosome.

## Results

### Sequential changes in the configuration of centriole engagement during daughter centriole maturation

A daughter centriole emerges from the wall of the mother centriole in early S phase and remains engaged until the next G1 phase. While engaged with the mother centriole, the daughter gradually changes its structure. To examine whether the configuration of centriole engagement also changes during this maturation process, we used Ultrastructure Expansion Microscopy (U-ExM) to obtain high-resolution images of the centriole pairs (Gambarotto et al, 2019). During interphase, most mother–daughter centriole pairs exhibited orthogonal engagement (Fig. 1A,B), while a smaller fraction displayed acute or obtuse engagement angles. As cells progressed into mitosis, the engagement angle shifted towards obtuse (Fig. 1A,B). The engagement dissociated during the subsequent G1 phase, as indicated by a >750 nm separation between mother and daughter CP110 signals (Ito et al, 2021). Notably, the daughter centriole gradually expands in width through the cell cycle (Fig. 1A,C). In early S phase, the daughter centriole's width measured $141 \pm 25$ nm, which increased to $167 \pm 26$ nm in late S phase and to $190 \pm 18$ nm by mitosis, reaching the width of the mother centriole ($189 \pm 19$ nm). These results are consistent with a recent study showing that daughter centriole width expands from around 140 to 195 nm during the bloom phase after duplication (Laporte et al, 2024). For clarity, we refer to centrioles with a width of less than 160 nm as "thin" in the following sections. Despite changes in both angle and width, the daughter centriole remained engaged with the mother centriole, indicating that centriole engagement undergoes dynamic reconfiguration throughout the daughter centriole's maturation.

Early S phase centrioles were thinner and had more variability in angle, with a greater proportion of non-orthogonal alignments compared to late S phase (Fig. 1B,C). To confirm this observation, we arrested cells in early S phase using Hydroxyurea (HU, a DNA replication inhibitor) and analyzed the centriole pairs (Fig. 1D). While the daughter centrioles remained thin in HU-treated cells (Fig. 1D,F; $135 \pm 13$ nm; Fig. EV1D,E), they maintained an orthogonal engagement angle with their mother centrioles, suggesting that the non-orthogonal configuration occurs before the HU arrest point (Fig. 1D,E). To test this hypothesis, we treated cells with a CDK2 inhibitor in conjunction with HU, as CDK2 is thought to regulate cell cycle progression from late G1 to S phase and has been implicated in centriole/centrosome formation (Tsai et al, 1993; Matsumoto et al, 1999; Hinchcliffe et al, 1999; Duensing et al, 2007). Immunostaining for phosphorylated Rb protein and FACS analysis confirmed that CDK2 inhibition with HU treatment arrested the cell cycle earlier than HU treatment alone (Fig. EV1A–C). This condition not only resulted in a decrease in daughter centriole width (Figs. 1D,F and EV1F: $139 \pm 18$ nm) but also an increase in the fraction of centrioles exhibiting acute or obtuse engagement angles with their mothers (Fig. 1D,E). These results suggest that the daughter centriole is thin with a flexible orientation immediately after duplication. As the cell cycle progresses through S phase, it transitions to a stable, orthogonal alignment and subsequently blooms.

Overall, these observations demonstrate the existence of at least four distinct configurations of mother–daughter centriole engagement (Fig. 1I). Stage 1, observed immediately after duplication, is characterized by a thin daughter centriole and a flexible association with the mother centriole. Stage 2 occurs in early S phase, where the width of the daughter centriole remains thin but establishes a more stable, orthogonal link with the mother. During late S and G2 phases, the centriolar configuration transitions to Stage 3, characterized by the initiation of daughter centriole blooming and continued orthogonal association with the mother. Finally, in Stage 4, which occurs during mitosis, the majority of centriole pairs exhibit an obtuse engagement angle. Treatment with HU + CDK2 inhibitor specifically enriches for the transient configurations of Stage 1, while HU treatment alone enriches for Stage 2. Untreated cells display a mixture of engagement profiles characteristic of Stage 1 and Stage 2 in early S phase, while predominantly exhibiting Stage 3 or Stage 4 profiles in late S/G2 and mitosis, respectively.

### Multiple mechanisms contribute to centriole engagement during interphase

Our previous work demonstrated that suppressing the expression of both Cep57 and Cep57L1 leads to precocious centriole disengagement in interphase, often followed by centriole re-duplication (Fig. EV1G,H) (Ito et al, 2021). This finding was further confirmed using knockout cells (Fig. EV2A–E). To determine whether Cep57 and Cep57L1 maintain all three interphase centriole engagement configurations (Stages 1–3), we knocked down their expression and assessed centriole behavior under each condition. Depletion of Cep57 and Cep57L1 resulted in precocious centriole disengagement and reduplication in cycling late S phase (Stage 3) cells, which was not observed in early S cells treated with HU + CDK2 inhibitor (Stage 1) or HU alone (Stage 2) (Fig. 1G,H). Supporting this observation, co-depletion of Cep57

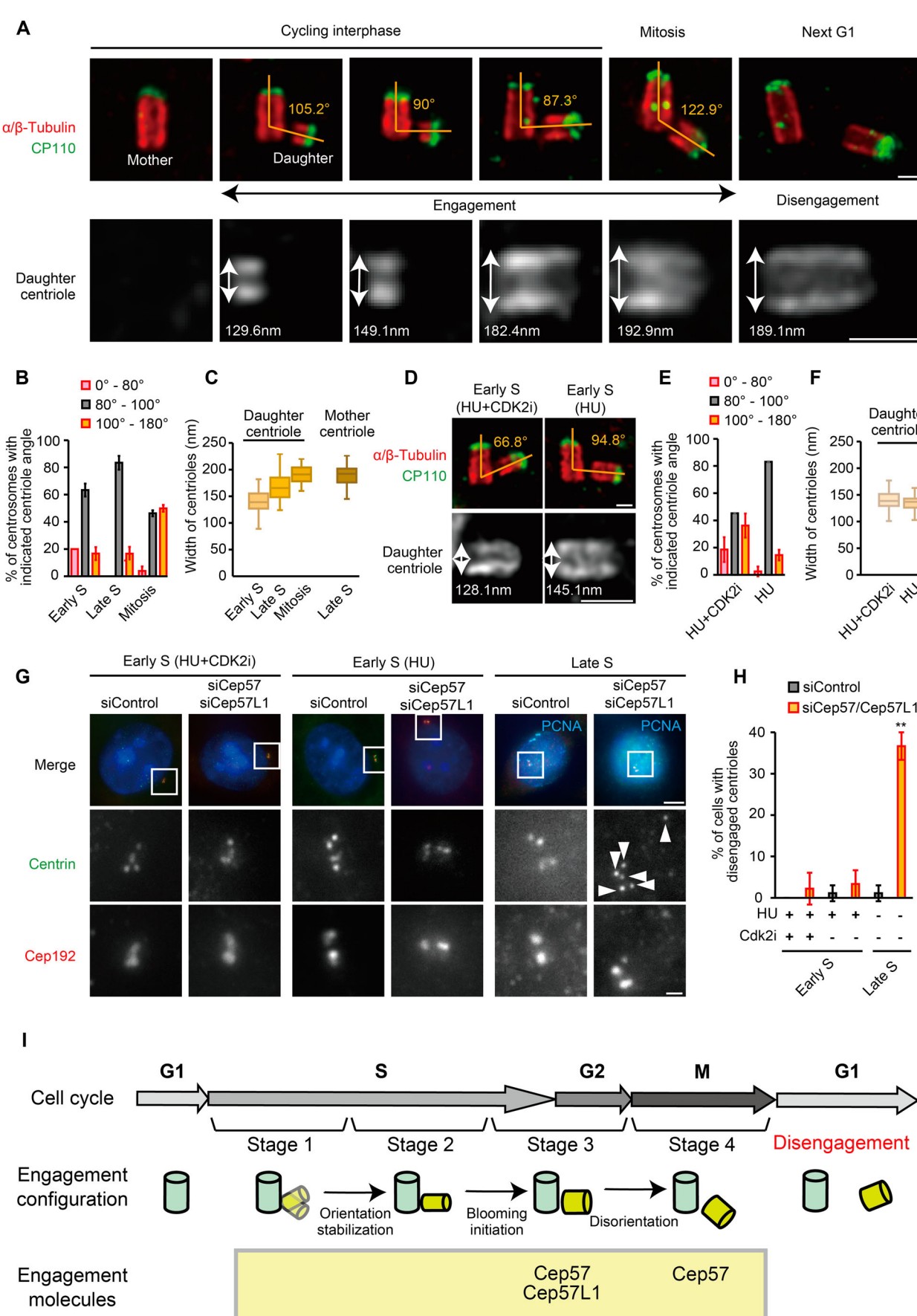

◄ **Figure 1.  Distinct stages of centriole engagement.**

(A) Ultrastructure expansion microscopy (U-ExM) images of centrioles through the cell cycle in HeLa cells. Scale bars: 200 nm. Double-headed arrows indicate the width of centrioles. (B) Quantification of the frequency of centrosomes with indicated mother–daughter centriole angles (0–80°, 80–100°, and 100–180°) observed by STED microscopy. The cell cycle phase was determined by the signal pattern of PCNA. Error bars represent the mean ± s.d. of three biological independent experiments. (C) Quantification of the widths of daughter centrioles observed by STED microscopy. The cell cycle phase was determined by the signal pattern of PCNA. Boxplot was employed to illustrate the data distribution. The central line represents the median, the box spans the interquartile range (IQR) from the 25th to the 75th percentile, and the whiskers extend to the smallest and largest data points within 1.5 times the IQR. (D) U-ExM images of centrioles in HeLa cells treated with hydroxyurea (HU) or HU + CDK2 inhibitor III (CDK2i). Scale bars: 200 nm. (E) Quantification of the frequency of centrosomes with indicated mother–daughter centriole angles (0–80°, 80–100°, and 100–180°) observed by STED microscopy. The cell cycle phase was determined by the signal pattern of PCNA. Error bars represent the mean ± s.d. of three biological independent experiments. (F) Quantification of the widths of daughter centrioles observed by STED microscopy. The cell cycle phase was determined by the signal pattern of PCNA. Boxplot was employed to illustrate the data distribution. The central line represents the median, the box spans the interquartile range (IQR) from the 25th to the 75th percentile, and the whiskers extend to the smallest and largest data points within 1.5 times the IQR. (G) Representative immunofluorescence images of HeLa cells transfected with siControl or siCep57/Cep57L1 under deionized distilled water (DDW), HU, or HU + CDK2i treatment. Scale bars: 5 μm, 1 μm. White arrowheads indicate the centrioles. (H) Quantification of the frequency of cells with disengaged centrioles in (G). Error bars represent the mean ± s.d. of three biological independent experiments. P values were calculated by two-tailed unpaired Welch's t-test with Bonferroni correction (P = 0.0011). **P < 0.01. (I) Schematic model of the four stages of centriole engagement throughout the cell cycle. Stage 1: flexible intercentriolar link and thin daughter centrioles. Stage 2: orthogonal intercentriolar link and thin daughter centrioles. Stage 3: expanded daughter centrioles, engaged to the mother by Cep57-Cep57L1. Stage 4: daughter centrioles at an obtuse angle, engaged to the mother solely by Cep57. Centrioles disengage in the following G1 phase. Source data are available online for this figure.

and Cep57L1 did not induce centriole disengagement in cycling early S phase (Stage 1 or 2) (Fig. EV2F,G). Therefore, centriole engagement during early S phase (Stages 1 and 2) is maintained through mechanisms distinct from the canonical Cep57/Cep57L1-dependent pathway during late S and G2 phases (Stage 3). Moreover, previous studies have shown that Cep57 depletion alone induces centriole disengagement in mitosis (Stage 4, Fig. EV1G,I) (Watanabe et al, 2019; Aziz et al, 2018). Collectively, these results indicate that multiple mechanisms ensure centriole engagement throughout the cell cycle (Fig. 1I).

## The cartwheel structure contributes to centriole engagement in early S phase: Stages 1 and 2

As Stage 1 occurs immediately after the onset of centriole duplication in early S phase, we hypothesized that the cartwheel, a key structure in this process, might contribute to centriole engagement during this stage. This hypothesis is supported by a previous study that revealed the connection between the cartwheel and the mother centriole wall through cryo-electron microscopy (Guichard et al, 2010). To determine whether the cartwheel contributes to centriole engagement, we aimed to remove it after centriole duplication. Given that the maintenance of the cartwheel depends on persistent PLK4 activity (Kim et al, 2016), the PLK4 inhibitor centrinone was added after daughter centriole formation. Cells were first arrested in early S phase by treatment with HU + CDK2 inhibitor for 24 h, inducing the Stage 1 state. At this point, all cells displayed newly formed daughter centrioles with cartwheels, associated with their respective mothers, indicating that CDK2 activity is not required for the initial formation of daughter centrioles (Fig. 2A,B, HU + CDK2i: 100.0%). Subsequent addition of the PLK4 inhibitor for 24 h resulted in cartwheel loss in approximately half of the cells (44.4 ± 15.0%). Unexpectedly, in Stage 1 cells lacking the cartwheel, the number of centrioles decreased from four to two in about 90% of cells (Fig. 2A,B, ≧3 centrioles: 11.1 ± 11.1%), suggesting a defect in daughter centriole integrity. Interestingly, among the cells retaining four centrioles after cartwheel loss, the minor population exhibited centriole disengagement (Fig. EV2H). This suggests that daughter centrioles are engaged with its mother through the cartwheel structure at this

stage, and that disengaged daughter centrioles are unstable, undergoing rapid disintegration.

In contrast, Stage 2 cells retained four centrioles even after cartwheel loss, and mother–daughter centriole engagement was maintained, indicating that cartwheel loss does not affect engagement stability (Fig. 2B–D, Disengagement: 4.4 ± 1.9%). We hypothesized that Cep57 and Cep57L1, which maintain centriole engagement in Stage 3, might also function redundantly with the cartwheel to maintain centriole engagement in Stage 2. To test this hypothesis, the cartwheel was eliminated in Cep57 and Cep57L1-depleted Stage 2 cells (Fig. 2C,D). Remarkably, precocious centriole disengagement was observed under this condition (Fig. 2C,D, 30.9 ± 7.9%). Cells retaining the cartwheel after PLK4 inhibitor treatment did not exhibit precocious centriole disengagement, indicating that the cartwheel, rather than PLK4 activity itself, is required for engagement (Figs. 2D, 3.3 ± 3.3%). These findings demonstrate that Cep57, Cep57L1, and the cartwheel redundantly maintain centriole engagement during Stage 2.

## CDK2-cyclin E establishes orthogonal centriole engagement: transition from Stage 1 to Stage 2

CDK2 inhibition leads to an accumulation of centrioles in Stage 1, suggesting that CDK2 activity is required for the transition to Stage 2. CDK2 is activated by forming complexes with either cyclin E or cyclin A (Malumbres, 2014). Immunofluorescence analysis revealed that cyclin E expression increases during late G1/early S phase (Fig. 2E), while cyclin A expression rises during late S phase (Fig. 3A). This temporal expression pattern implies that CDK2-cyclin E activity may be responsible for initiating the transition from Stage 1 to Stage 2. To identify which cyclin-CDK2 complex mediates this transition, we depleted each cyclin and examined centriole configurations during S phase. Control cells exhibited orthogonal centrioles, whereas cyclin E depletion, but not cyclin A depletion, resulted in unstable centriole angles (Fig. 2F,G). This suggests that CDK2-cyclin E regulates the transition from Stage 1 to Stage 2 centriole configuration. The mechanism of centriole engagement also differs between Stages 1 and 2, with Stage 1 relying on the cartwheel structure alone and Stage 2 transitioning to a cartwheel-Cep57-Cep57L1-dependent state. Similar to CDK2

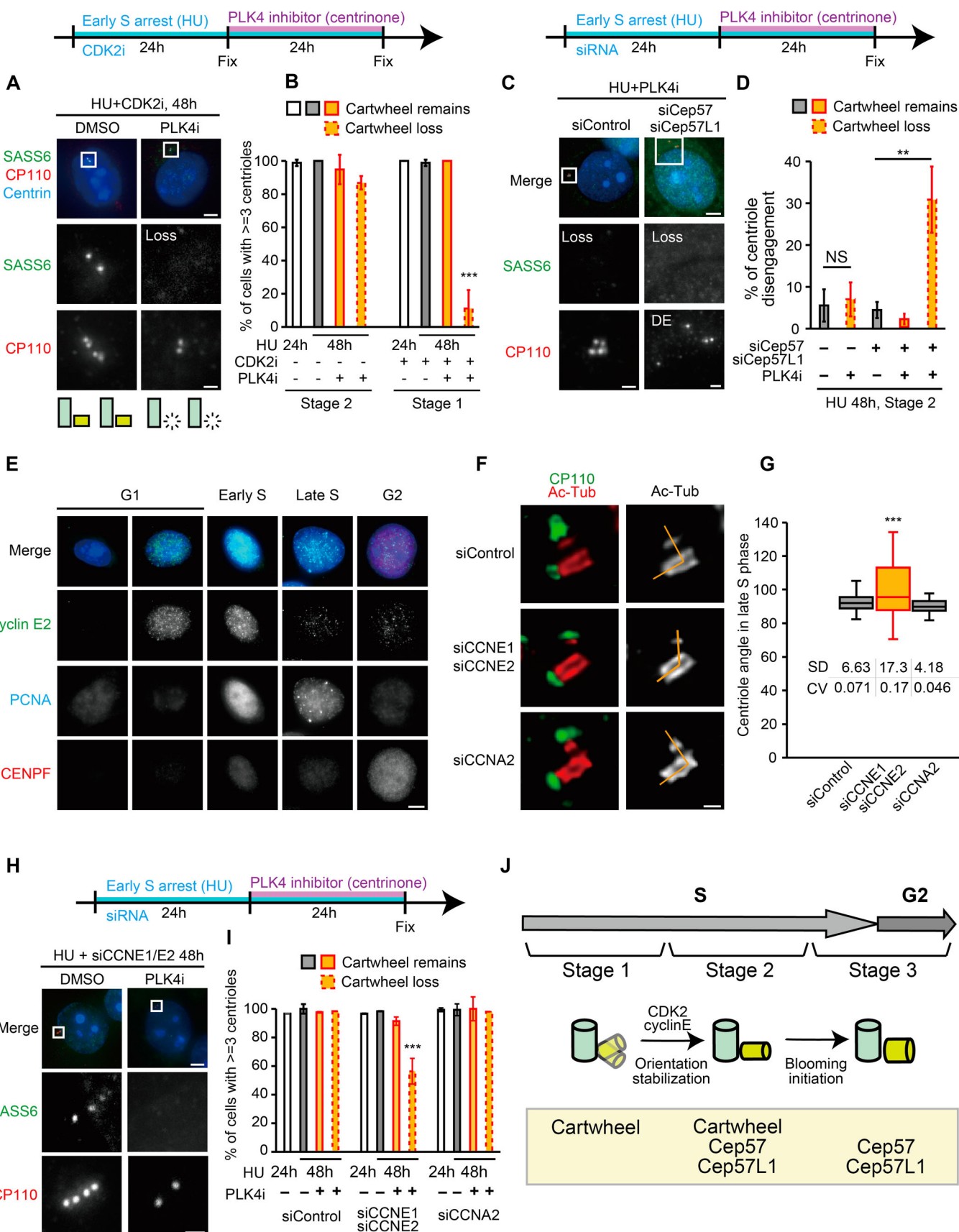

**Figure 2.  The cartwheel contributes to centriole engagement in early S phase.**

(**A**) Representative immunofluorescence images of HeLa cells subjected to a cartwheel removal assay. HeLa cells were treated with HU and CDK2 inhibitor III (CDK2i) for 24 h, followed by treatment with PLK4 inhibitor for another 24 h. Scale bars: 5 μm, 1 μm. (**B**) Quantification of the frequency of cells containing three or more centrioles in (**A**). Solid bars indicate the centrosomes that retain the cartwheel structure, while dashed bars represent the centrosomes that have lost the cartwheel structure. Error bars represent the mean ± s.d. of three biological independent experiments. *P* values were calculated by Dunnett's multiple comparisons test (*P* = 4.87E−09). ***P* < 0.001. (**C**) Representative immunofluorescence images of HeLa cells subjected to a cartwheel removal assay. HeLa cells were treated with HU and transfected with indicated siRNAs for 24 h, followed by treatment with a PLK4 inhibitor for another 24 h. Scale bars: 5 μm, 1 μm. (**D**) Quantification of the frequency of cells with disengaged centrioles in (**C**). Error bars represent the mean ± s.d. of three biological independent experiments. *P* values were calculated by two-tailed unpaired Welch's t-test (siControl, *P* = 0.686) and Dunnett's multiple comparisons test (siCep57 + siCep57L1, *P* = 0.00321). NS, not significant; ***P* < 0.01. (**E**) Immunofluorescence images of HeLa cells throughout the cell cycle. The cell cycle phase was determined by the signal pattern of PCNA and CENP-F. Scale bar: 5 μm. (**F**) Representative STED microscopy images of centrosomes in HeLa cells in late S phase transfected with indicated siRNAs. Scale bar: 200 nm. (**G**) Quantification of the frequency of centrosomes with the indicated mother–daughter centriole angles in (**F**). Boxplot was employed to illustrate the data distribution. The central line represents the median, the box spans the interquartile range (IQR) from the 25th to the 75th percentile, and the whiskers extend to the smallest and largest data points within 1.5 times the IQR. SD: standard deviation; CV: coefficient of variation. *P* values were calculated by F test with Bonferroni correction (*P* = 7.33E−06). ***P* < 0.001. (**H**) Representative immunofluorescence images of HeLa cells subjected to a cartwheel removal assay. HeLa cells were treated with HU and siRNAs for 24 h, followed by treatment with a PLK4 inhibitor for another 24 h. Scale bars: 5 μm, 1 μm. (**I**) Quantification of the frequency of cells containing three or more centrioles in (**H**). Solid bars indicate the centrosomes that retain the cartwheel structure, while dashed bars represent the centrosomes that have lost the cartwheel structure. Error bars represent the mean ± s.d. of three biological independent experiments. *P* values were calculated by Dunnett's multiple comparisons test (*P* = 7.55E−04). ***P* < 0.001. (**J**) Schematic model of the mechanisms and configurations of centriole engagement during interphase. Source data are available online for this figure.

inhibition (Fig. 2A,B), cartwheel removal by PLK4 inhibitor led to centriole disengagement and loss of daughter centrioles in cyclin E-depleted cells (Fig. 2H,I; 4 centrioles: 56.4 ± 8.9%). These findings indicate that CDK2-cyclin E mediates the transition in both the configuration and mechanism of centriole engagement from Stage 1 to Stage 2 (Fig. 2J).

## Centriole blooming reduces reliance on the cartwheel for maintaining engagement: transition from Stage 2 to Stage 3

During late S phase, the configuration and mechanism of centriole engagement change: daughter centrioles undergo blooming, and the engagement mechanism transitions from a cartwheel-Cep57-Cep57L1-dependent state (Stage 2) to a Cep57-Cep57L1-dependent state (Stage 3) (Fig. 2J). In other words, the cartwheel becomes dispensable for maintaining engagement. During this phase, cyclin A, which forms a complex with CDK2 (Malumbres, 2014), begins to accumulate (Fig. 3A). We therefore investigated whether the CDK2-cyclin A complex drives the transition from Stage 2 to Stage 3. In Cep57L1 knockout cells, Cep57 depletion led to precocious centriole disengagement in late S phase, whereas simultaneous cyclin A depletion reduced this phenotype (Fig. 3B,C; siCep57: 55.0 ± 4.2%, siCep57+siCCNA2: 21.1% ± 7.1%). CDK2 inhibitor also suppressed the precocious centriole disengagement in Cep57/Cep57L1-depleted cells (Fig. 3D, DMSO: 62.7 ± 9.5%, CDK2i: 17.3 ± 5.7%). STED microscopy analysis revealed that cyclin A depletion resulted in daughter centrioles that remained as thin as those observed in early S phase (Fig. 3E,F; siControl: 165 ± 20 nm, siCCNA2: 127 ± 22 nm), indicating daughter centriole blooming did not initiate. These findings suggest that CDK2-cyclin A complex mediates the transition in both centriole engagement configuration and mechanism from Stage 2 to Stage 3.

To investigate the relationship between daughter centriole blooming and changes in the engagement mechanism, we sought to suppress blooming by depleting Cep295 or CPAP. While both Cep295 and CPAP have been implicated in daughter centriole maturation (Izquierdo et al, 2014; Chang et al, 2016), their specific roles in centriole blooming were previously unknown. Depletion of

either Cep295 or CPAP indeed resulted in thinner daughter centrioles even in late S phase, indicating their crucial role in blooming (Fig. 3E,F; siCep295: 126 ± 24 nm; Fig. EV2I,J, siCPAP: 142 ± 13 nm). When blooming was suppressed by depleting Cep295 or CPAP, the frequency of precocious centriole disengagement in Cep57-depleted Cep57L1 knockout cells was reduced (Figs. 3G,H and EV2K,L). These results suggest that daughter centriole blooming, facilitated by both Cep295 and CPAP, drives the transition of the centriole engagement mechanism from Stage 2, where the engagement relies on the cartwheel, Cep57, and Cep57L1, to Stage 3, where only Cep57 and Cep57L1 are required. This indicates that daughter centriole blooming renders the cartwheel dispensable for centriole engagement following this transition. U-ExM analysis revealed that the distance between the cartwheel and the microtubule wall increased during daughter centriole blooming (Fig. 3I,J). Thus, we speculate that daughter centriole blooming weakens the connection between the daughter centriole wall and its internal cartwheel (Fig. 3K).

## The torus and PCM surrounding mother centrioles maintain engagement during late S to G2 phase: Stage 3

During Stage 3 (late S phase to G2 phase), centriole engagement is maintained by Cep57 and Cep57L1 around the mother centriole wall (Figs. 1 and EV1) (Ito et al, 2021). However, STED microscopy revealed a gap between the outer edge of the Cep57/Cep57L1 ring (radius: 135 nm) and the proximal end of the daughter centriole (approximately 160 nm from the mother centriole's center) (Fig. 4A,B). This gap suggests that additional linking factors are involved in maintaining centriole engagement. To identify these additional factors, we investigated the roles of Cep63, Cep152, and PCNT, which interact with Cep57/Cep57L1 and are localized further from the mother centriole's center (Fig. 4A,B) (Wei et al, 2020; Watanabe et al, 2019; Zhao et al, 2020; Lukinavičius et al, 2013). Individual siRNA knockdowns of these proteins did not induce precocious centriole disengagement during G2 phase (Fig. 4D), indicating possible redundancy in their functions. To explore this redundancy, we performed co-depletion screens targeting combinations of Cep57, Cep57L1, Cep63, Cep152, and

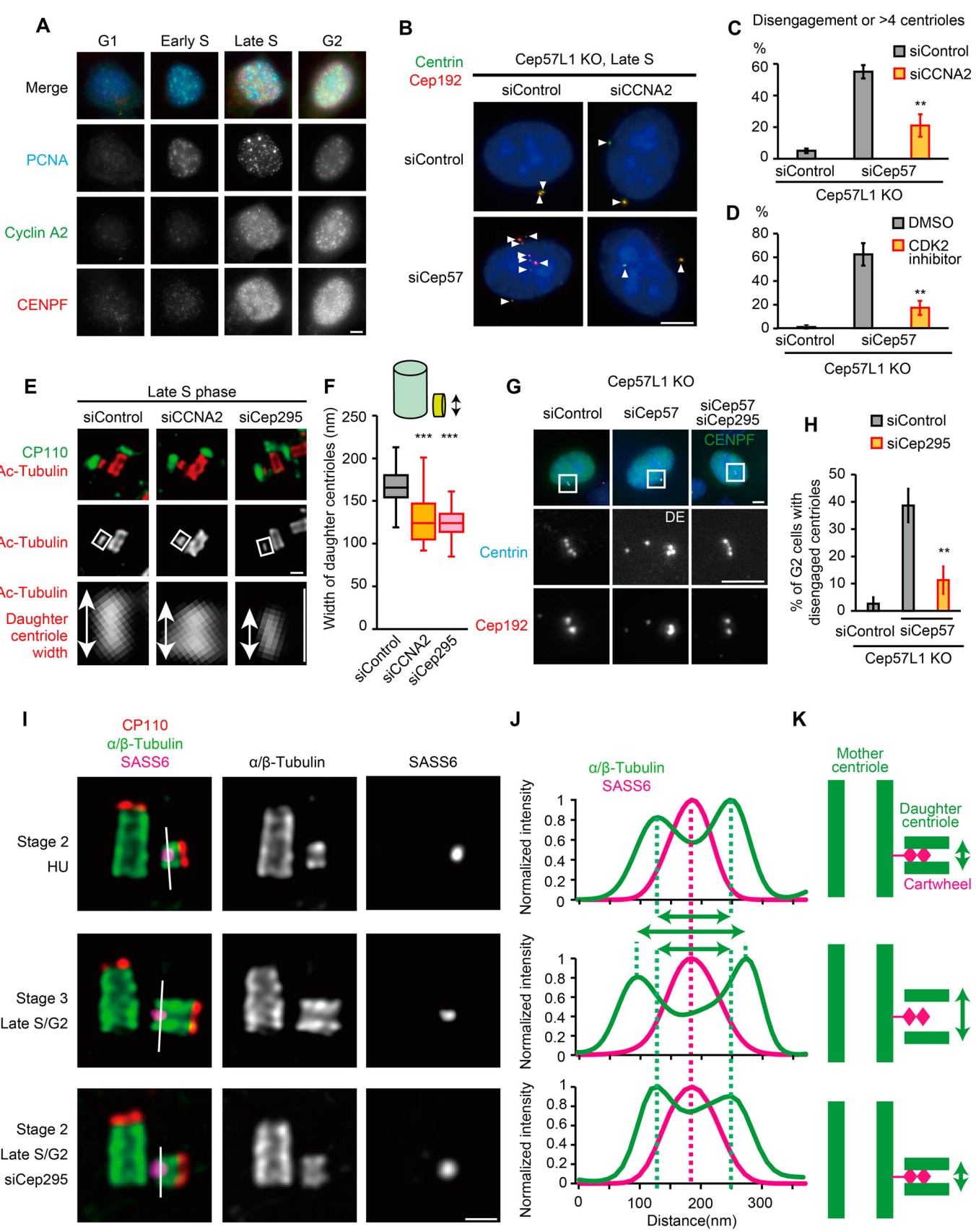

**Figure 3. Daughter centriole blooming releases the cartwheel centriole engagement pathway.**

(A) Immunofluorescence images of HeLa cells throughout the cell cycle. The cell cycle phase was determined by the signal pattern of PCNA and CENP-F. Scale bar: 5 µm. (B) Representative immunofluorescence images of HeLa Cep57L1KO cells transfected with siControl and siCep57, in the presence or absence of siCCNA2, Scale bar: 5 µm. White arrowheads indicate the centrioles. (C) Quantification of the frequency of cells with disengaged centrioles in (B). The cell cycle phase was determined by the signal pattern of PCNA. Error bars represent the mean ± s.d. of three biological independent experiments. *P* value was calculated by two-tailed unpaired Welch's t-test (*P* = 0.00164). **P < 0.01. (D) Quantification of the frequency of cells with disengaged centrioles in late S phase of Cep57L1KO cells transfected with siControl and siCep57, in the presence or absence of CDK2 inhibitor. The cell cycle phase was determined by the signal pattern of PCNA. Error bars represent the mean ± s.d. of three biological independent experiments. *P* value was calculated by two-tailed unpaired Welch's t-test (*P* = 0.00437). **P < 0.01. (E) Representative STED microscopy images of HeLa cells transfected with siControl, siCCNA2, or siCep295. Double-headed arrows indicate the width of centrioles. Scale bar: 200 nm. (F) Quantification of the width of the daughter centrioles. Boxplot was employed to illustrate the data distribution. The central line represents the median, the box spans the interquartile range (IQR) from the 25th to the 75th percentile, and the whiskers extend to the smallest and largest data points within 1.5 times the IQR. *P* value was calculated by two-tailed unpaired Welch's t-test with Bonferroni correction (siCCNA2, *P* = 2.71E−07; siCep295, *P* = 4.83E−10). ***P < 0.001. (G) Representative immunofluorescence images of HeLa Cep57L1 KO cells transfected with siControl, siCep57, or siCep57 + siCep295. Scale bars: 5 µm. (H) Quantification of the frequency of cells with disengaged centrioles in (G). Error bars represent the mean ± s.d. of three biological independent experiments. *P* value was calculated by two-tailed unpaired Welch's t-test (*P* = 0.00440). **P < 0.01. (I) U-ExM images of centrioles in HeLa cells treated with HU (early S), DDW (late S or G2), or siCep295 (late S or G2). Scale bar: 200 nm. (J) Quantification of the signal intensity of α/β-tubulin and SASS6 in (I). (K) Schematic model for the release of the cartwheel engagement by daughter centriole blooming. Source data are available online for this figure.

PCNT. Only three co-depletion combinations resulted in precocious centriole disengagement: siCep57 + siCep57L1, siCep57 + siCep63, and siCep63 + siPCNT (Fig. 4C,D; siControl: 2.0 ± 2.7%, siCep57+siCep57L1: 64.3 ± 0.5%, siCep57+siCep63: 47.2 ± 2.6%, siCep63+siPCNT: 33.7 ± 3.3%). While PCNT is known to maintain centriole engagement during mitosis (Lee and Rhee, 2012; Pagan et al, 2015; Matsuo et al, 2012), the role of Cep63 in this process has not been previously reported. To validate Cep63's involvement, we generated a Cep63 knockout cell line using CRISPR-Cas9 (Fig. EV3A–C). Contrary to earlier studies reporting multipolar spindle formation and reduced centriole numbers in Cep63 knockout cells (Brown and Costanzo, 2009; Brown et al, 2013), our generated cell line did not exhibit these phenotypes (Fig. EV3B,D,E). Importantly, treatment of Cep63 knockout cells with siCep57 or siPCNT induced centriole disengagement in interphase, aligning with our co-depletion results (Figs. 4C,D and EV3F,G; siControl: 2.2 ± 1.9%, siCep57: 41.2 ± 8.4%, siPCNT: 35.9 ± 3.0%). These findings demonstrate that both Cep63 and PCNT, in addition to Cep57 and Cep57L1, contribute to centriole engagement during the late S to G2 phase (Stage 3).

While PCNT also maintains centriole engagement during mitosis with Cep57 (Watanabe et al, 2019), Cep63, along with Cep57L1, appears to be an interphase-specific factor, as Cep63 knockout does not induce centriole disengagement in mitosis (Fig. EV3H). The result that simultaneous depletion of Cep63 with either PCNT or Cep57 led to centriole disengagement suggests that two pathways support interphase engagement: an interphase-specific Cep63/Cep57L1 pathway and a Cep57/PCNT pathway that also functions during mitosis. During interphase, Cep63 and Cep57L1 interact and localize in close proximity, with the Cep63 ring encircling Cep57L1 on the outer surface of the mother centriole (Figs. 4A and EV3I–K) (Zhao et al, 2020). We hypothesized that Cep57L1 might recruit Cep63 to the engagement site. However, Cep57L1 depletion did not affect Cep63 signal intensity at the centrosome (Fig. 4E,F). Notably, simultaneous depletion of Cep57 and Cep57L1 significantly reduced Cep63 intensity (Fig. 4E,F) (Ito et al, 2021; Zhao et al, 2020). These observations suggest that within the interphase-specific pathway, Cep57 and Cep57L1 redundantly regulate centriole engagement through Cep63. This redundancy explains why co-depletion of Cep57 + PCNT or Cep57L1 + PCNT did not lead to premature

disengagement: the remaining protein (Cep57L1 or Cep57, respectively) could still maintain engagement via Cep63. In summary, during late S to G2 phase (Stage 3), two pathways maintain centriole engagement: the Cep57-PCNT pathway and the Cep57-Cep57L1-Cep63 pathway. Since the outermost proteins in each pathway, PCNT and Cep63, are components of the PCM (Pimenta-Marques and Bettencourt-Dias, 2020) and the torus (Lukinavičius et al, 2013), respectively, we designate the Cep57-PCNT as the PCM pathway and the Cep57-Cep57L1-Cep63 as the torus pathway (Fig. 4G).

## The torus protein Cep63 ensures the orthogonal engagement of mother–daughter centrioles

Remarkably, Cep63 knockout cells exhibited a loss of orthogonal engagement between mother–daughter centrioles during late S phase, with an increase in obtuse centriole angles (Fig. 4H,I; obtuse centriole angle, HeLa-Cas9: 16.6 ± 5.0%, Cep63KO: 42.2 ± 10.8%). The obtuse engagement was also observed in HU-arrested early S phase, indicating that stable orthogonal engagement fails to form in Cep63KO cells (Fig. 4H,I). Obtuse centriole angles are typically observed during mitosis (Stage 4), suggesting that the torus pathway reduces its contribution to maintaining engagement during the transition from Stage 3 to Stage 4. Depletion of other torus or PCM proteins, which do not disrupt the torus pathway, did not lead to an increase in obtuse centriole angles (Fig. 4J,K). Therefore, the torus pathway is crucial for the orthogonal engagement between mother–daughter centrioles.

## Centriole distancing mediated by the loss of the daughter's proximal end drives the transition of the centriole engagement mechanism from Stage 3 to Stage 4

The obtuse configuration observed in early mitosis is reportedly caused by centriole distancing, a process in which the distance between the mother centriole wall and the proximal end of the daughter centriole slightly increases (Shukla et al, 2015). This phenomenon is mediated by the kinase PLK1 (Fig. 5A,B) (Shukla et al, 2015). We hypothesized that centriole distancing also alters the centriole engagement mechanism from Stage 3 to Stage 4.

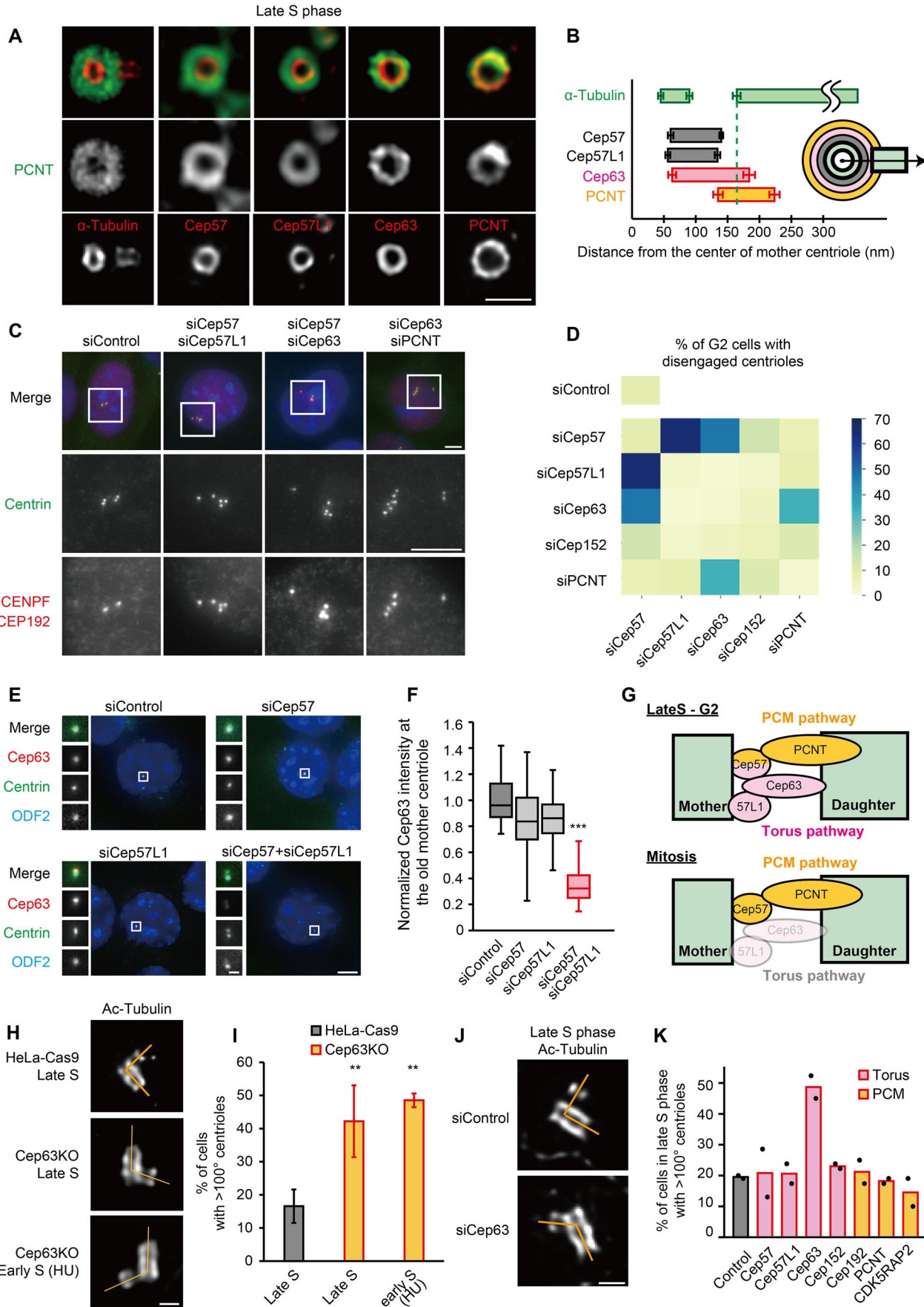

Figure 4. Centriole engagement is maintained redundantly by the torus and the PCM pathways from late S to G2 phase.

(A) Representative STED microscopy images of centrioles and surrounding centrosomal proteins in HeLa cells. The cell cycle phase was determined by the signal pattern of PCNA. Scale bar: 200 nm. (B) Quantification of the radial distribution of the indicated centrosomal proteins relative to the center of the mother centriole. The proximal and distal ends of the signal were defined as the location along the radial axis where the signal intensity dropped to 20% of its peak value. Data are presented as horizontal bars representing the mean radial distance of the signal from the center of the mother centriole, with error bars indicating the s.d. $n = 30$ centrioles pooled from 3 independent experiments. (C) Representative immunofluorescence images of HeLa cells transfected with the indicated siRNAs. Scale bars: 5 μm. (D) Quantification of the frequency of cells with disengaged centrioles in (C). $n = 2$ independent experiments, 50 cells each. (E) Representative immunofluorescence images of HeLa cells transfected with the indicated siRNAs. Scale bars: 5 μm, 1 μm. (F) Quantification of signal intensity of Cep63 at the old mother centriole in (E). Boxplot was employed to illustrate the data distribution. The central line represents the median, the box spans the interquartile range (IQR) from the 25th to the 75th percentile, and the whiskers extend to the smallest and largest data points within 1.5 times the IQR. $P$ value was calculated by Dunnett's multiple comparisons test ($P < 1E-30$). ***$P < 0.001$. (G) Schematic model for the molecular mechanisms of centriole engagement. (H) Representative STED microscopy images of centrioles in late S phase of HeLa-Cas9, late S phase of Cep63KO HeLa-Cas9, and early S phase of Cep63KO HeLa-Cas9 cells. Scale bar: 200 nm. (I) Quantification of the frequency of the centrosomes, in which the centrioles are connected at an obtuse angle in (H). Error bars represent the mean ± s.d. of three biological independent experiments. $P$ values were calculated by Dunnett's multiple comparisons test (Cep63KO Late S, $P = 0.00720$; Early S, $P = 0.00209$). **$P < 0.01$. (J) Representative STED microscopy images of centrioles in late S phase upon indicated siRNA treatments. Scale bar: 200 nm. (K) Quantification of the frequency of the centrosomes, in which the centrioles are connected at an obtuse angle in (J). Two biologically independent experiments, 30 cells each. Source data are available online for this figure.

During mitosis (Stage 4), inhibition of the PCM pathway by Cep57 or PCNT depletion led to precocious centriole disengagement, while PLK1 inhibition prevented this disengagement (Fig. 5C,D). Furthermore, overexpression of constitutively active PLK1, which induces precocious centriole distancing (Fig. EV4A,B) (Shukla et al, 2015), caused precocious centriole disengagement upon Cep57 or PCNT depletion in interphase (Fig. EV4C,D). These results suggest that centriole distancing is both necessary and sufficient for the transition in the centriole engagement mechanism.

Closer examination of the centriole wall before and after centriole distancing revealed that the α/β-tubulin signal at the daughter centriole's proximal end is located further from the mother centriole during mitosis compared to interphase (Fig. 5E,F). In contrast, the acetylated tubulin signal, a marker of stabilized microtubules, was already distant from the mother centriole during interphase (Figs. 5E and EV4E,F). Notably, the gap between the acetylated tubulin and α/β-tubulin signals at the daughter's proximal end shortened in mitosis (Fig. 5E,G; Interphase: 27.1 ± 12.7 nm, Mitosis: 9.3 ± 12.2 nm). These observations suggest that the non-acetylated microtubules at the base of the daughter centriole are lost during mitotic entry. To further investigate this, we observed the structures located near the proximal end of the daughter centrioles; the cartwheel structure and the γ-tubulin ring complex (γ-TuRC) capping the A-tubule of the daughter centriole. The cartwheel was observed to be approximately 110 nm in length inside the daughter centriole during G2 phase, which shrank to around 90 nm in prometaphase (Fig. 5H,I). The γ-tubulin signal at the daughter's proximal end was reduced from the G2 phase to the next G1 phase, suggesting the loss of the γ-TuRC during mitosis (Fig. EV4G,H). Taken together, these findings strongly implicate the loss of the daughter centriole's proximal end during mitosis.

Interestingly, inhibition of PLK1, which prevents centriole distancing, also blocks the loss of non-acetylated microtubules, cartwheel shrinkage, and γ-TuRC dissociation at the daughter's proximal end (Figs. 5E–I and EV4G,H). These findings suggest that centriole distancing is caused by the loss of the daughter centriole's proximal end. To determine whether microtubule depolymerization from the minus end contributes to the loss of the proximal end, cells were treated with a high concentration (10 μM) of nocodazole. Although nocodazole typically destabilizes microtubules, high concentrations can stabilize their minus ends (Vasquez et al, 1997). Treatment with high-concentration nocodazole

suppressed the shrinkage of the non-acetylated microtubules at the daughter's proximal end during mitosis (Fig. 5E,G). This treatment also blocked the formation of an obtuse centriole angle (Fig. EV5A,B; >100° centrioles, DMSO: 49.9 ± 2.5%, Nocodazole: 23.5 ± 4.2%), and suppressed cartwheel shrinkage at mitotic entry (Fig. 5H,I). We also examined the effect on the cartwheel in prometaphase-arrested cells. Although the cartwheel is known to be degraded after metaphase (Strnad et al, 2007), our observations indicate that prolonged prometaphase arrest also leads to cartwheel loss (Fig. EV5C,D; STLC treatment: cartwheel retention, 18.9 ± 6.9%). High-dose nocodazole treatment, however, effectively suppressed this cartwheel loss in prometaphase-arrested cells (Fig. EV5C,D; 48.8 ± 15.7%). Note that the torus and PCM structures remained unaffected by this treatment, suggesting a specific effect on centriolar microtubules (Fig. EV5E,F). It is possible that PLK1 might cause the removal of the γ-tubulin cap, leading to microtubule depolymerization at the daughter's proximal end. Collectively, these data suggest that microtubule depolymerization induces the loss of the daughter centriole's proximal end, which subsequently triggers centriole distancing.

Next, we investigated the contribution of centriole distancing to the change in the centriole engagement mechanism during the transition from Stage 3 to Stage 4. To this end, cells were treated with high-concentration nocodazole, and mitotic centrioles were observed. This treatment prevented precocious centriole disengagement in mitosis induced by disrupting the PCM pathway through PCNT depletion (Fig. 5J,K, 10.0 ± 5.8%). In contrast, treatment with taxol or low-concentration nocodazole did not prevent precocious disengagement (Figs. 5J,K and EV5G,H). These results suggest that centriole distancing, driven by the loss of the daughter centriole's proximal end, promotes the transition of the centriole engagement mechanism from Stage 3 to Stage 4.

The α-tubulin signal at the proximal end overlapped with the ring of Cep63, whereas the acetylated tubulin signal, presumably of the mitotic daughter centriole, did not (Fig. 5L–N). In contrast, both signals overlapped with the ring of PCNT (Figs. 5L–N and EV4E). These observations suggest that the daughter centriole in the G2 phase contacts both the torus and the PCM, while the torus is separated in mitosis due to centriole distancing. This change in the configuration corresponds to the change in the mechanism of centriole engagement, whereby the PCM pathway alone becomes responsible for the engagement in mitosis. Collectively, we propose that centriole

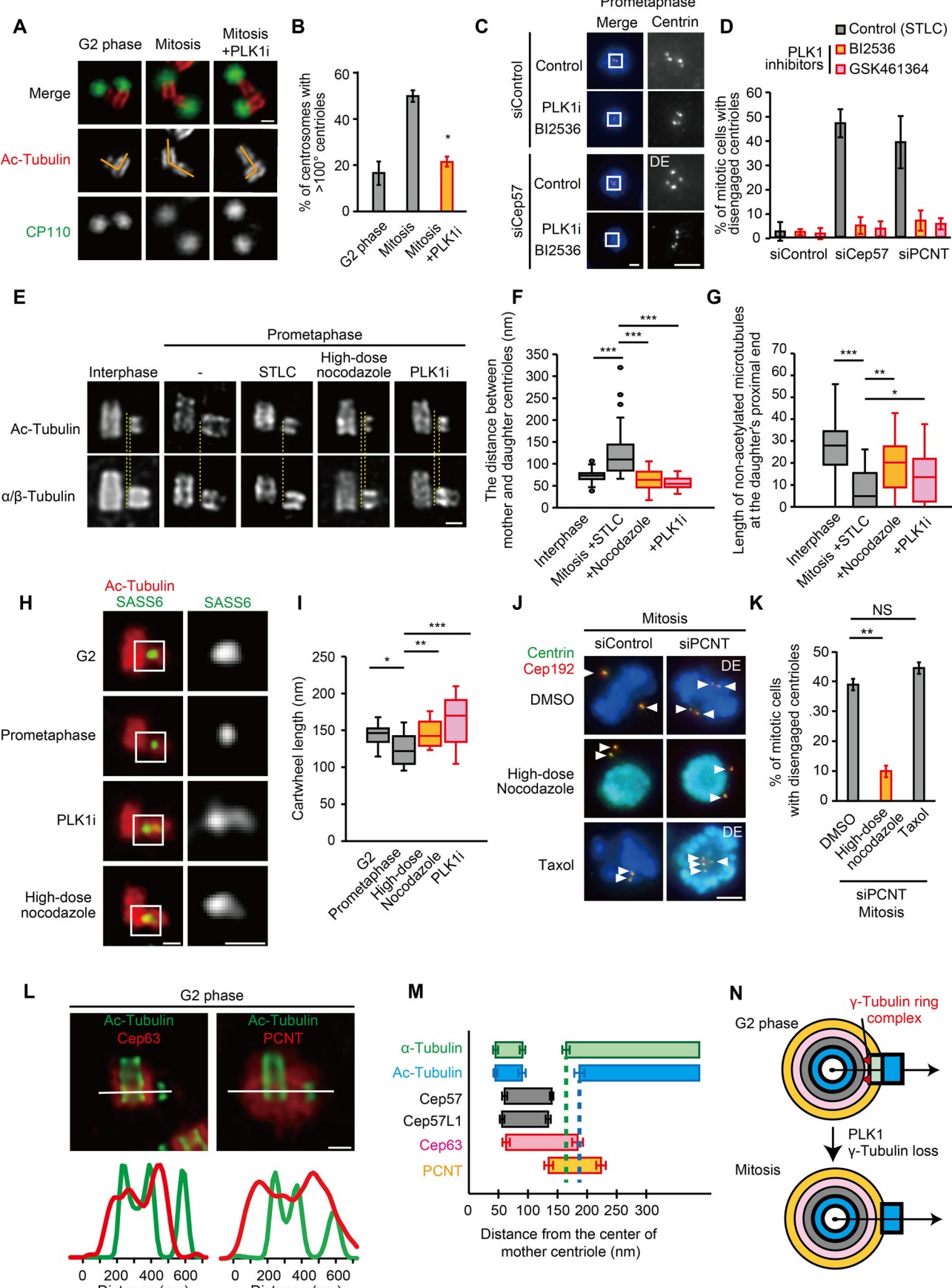

**Figure 5. Centriole distancing releases centriole engagement by the torus pathway.**

(A) Representative STED microscopy images of centrioles in G2 phase and mitosis treated with PLK1 inhibitor. Scale bar: 200 nm. (B) Quantification of the frequency of the centrosomes in which the centrioles are connected at an obtuse angle in (A). Error bars represent the mean ± s.d. of three biological independent experiments. $P$ value was calculated by two-tailed unpaired Welch's t-test ($P = 0.0202$). *$P < 0.05$. (C) Representative immunofluorescence images of HeLa cells in mitosis transfected with siControl and siCep57, in the presence or absence of PLK1 inhibitor (BI2536). Scale bars: 5 µm. (D) Quantification of the frequency of mitotic cells with disengaged centrioles in (C). Error bars represent the mean ± s.d. of three biological independent experiments. (E) Representative U-ExM images of centrioles in HeLa cells in interphase and mitosis. Scale bar: 200 nm. The yellow dotted line indicates the edges of the α-tubulin and acetylated-tubulin signals at the daughter's proximal end. (F) Quantification of the distance between mother and daughter centrioles in (E), measured by the α/β-tubulin signal. Boxplot was employed to illustrate the data distribution. The central line represents the median, the box spans the interquartile range (IQR) from the 25th to the 75th percentile, and the whiskers extend to the smallest and largest data points within 1.5 times the IQR. $P$ values were calculated by Dunnett's multiple comparisons test (interphase, $P = 2.58\text{E}{-}11$; Nocodazole, $P = 7.45\text{E}{-}12$; PLK1 inhibitor, $P = 1.40\text{E}{-}14$). ***$P < 0.001$. (G) Quantification of the length of non-acetylated microtubules at the daughter's proximal end in (E). Boxplot was employed to illustrate the data distribution. The central line represents the median, the box spans the interquartile range (IQR) from the 25th to the 75th percentile, and the whiskers extend to the smallest and largest data points within 1.5 times the IQR. $P$ values were calculated by Dunnett's multiple comparisons test (interphase, $P = 7.36\text{E}{-}09$; Nocodazole, $P = 0.00490$; PLK1 inhibitor, $P = 0.0414$). *$P < 0.05$, **$P < 0.01$, ***$P < 0.001$. (H) Representative U-ExM images of centrioles in HeLa cells. Scale bars: 200 nm. (I) Quantification of the length of the cartwheel in (H). Boxplot was employed to illustrate the data distribution. The central line represents the median, the box spans the interquartile range (IQR) from the 25th to the 75th percentile, and the whiskers extend to the smallest and largest data points within 1.5 times the IQR. $P$ values were calculated by Dunnett's multiple comparisons test (G2, $P = 0.0440$; Nocodazole, $P = 0.00147$; PLK1 inhibitor, $P = 0.0000513$). *$P < 0.05$, **$P < 0.01$, ***$P < 0.001$. (J) Representative immunofluorescence images of HeLa cells in mitosis transfected with siControl or siPCNT, in the presence or absence of high-dose nocodazole or taxol. Scale bar: 5 µm. White arrowheads indicate the centrioles. (K) Quantification of the frequency of mitotic cells with disengaged centrioles in (J). Error bars represent the mean ± s.d. of three biological independent experiments. $P$ values were calculated by Dunnett's multiple comparisons test (Nocodazole, $P = 0.00130$; Taxol, $P = 0.430$). NS, not significant; **$P < 0.01$. (L) STED microscopy images of centrioles in G2 phase. Scale bar: 200 nm. (M) Quantification of the radial distribution of the indicated centrosomal proteins relative to the center of the mother centriole. The proximal and distal ends of the signal were defined as the location along the radial axis where the signal intensity dropped to 20% of its peak value. Data are presented as horizontal bars representing the mean radial distance of the signal from the center of the mother centriole, with error bars indicating the s.d. $n = 30$ centrioles pooled from 3 independent experiments. (N) Schematic model for the release of the torus engagement pathway by daughter centriole distancing. Source data are available online for this figure.

distancing causes the transition in the engagement mechanism by releasing the daughter centriole from the torus engagement pathway, leading to the transition from Stage 3 to Stage 4.

## Stepwise transitions in centriole engagement are essential for proper centriole disengagement and centrosome number control

Our findings demonstrate that centriole engagement undergoes distinct alterations at specific cell cycle stages. To investigate whether these alterations occur in a stepwise manner, with each transition dependent on the completion of the preceding one, we examined the consequences of blocking each transition of centriole engagement. In control cells, a thin daughter centriole forms at the wall of its mother at an unstable angle and then undergoes orthogonal alignment (early S phase), blooming (late S phase), distancing with an obtuse engagement angle (mitotic entry), and finally, disengagement (G1 phase) as the cell cycle progresses (Fig. 6A). When cyclin E depletion inhibited the establishment of orthogonal engagement in early S phase, daughter centriole blooming (late S phase) did not occur, leaving the centrioles in a thin, unstable configuration (Fig. 6A). Likewise, when cyclin A2 depletion inhibited daughter centriole blooming (late S phase), obtuse angle formation (mitotic entry) did not occur during mitosis (Fig. 6A). Moreover, the failure of centriole blooming (late S phase) also resulted in a defect in centriole disengagement (G1 phase) (Fig. 6A–C). Treatment with nocodazole or the PLK1 inhibitor, which blocks centriole distancing (mitotic entry), also prevented centriole disengagement (G1 phase) (Fig. 6A–C). Therefore, the centriole engagement configuration undergoes stepwise transitions throughout the cell cycle, and each transition is essential for eventual centriole disengagement.

The disengaged daughter centriole matures into an independent, functional centrosome during the subsequent G1 phase. We therefore investigated how inhibiting the stepwise engagement transitions affects centriole-to-centrosome conversion. Because some, but not all, treatments that block centriole engagement transitions also caused mitotic arrest, we added STLC to all conditions to arrest all cells in mitosis and then forced them into G1 phase using a CDK1 inhibitor (Tsou et al, 2009). As cytokinesis is inhibited under these conditions, post-mitotic cells inherit both centrosomes that comprise the mitotic spindle poles, each containing a mother and a daughter centriole. During the subsequent G1 phase, these centrioles disengage, and the now-independent daughter centrioles convert into centrosomes. Consequently, control cells end up with more than two centrosomes (Fig. 6B,D). In contrast, inhibiting any of the centriole engagement transitions reduced the number of converted centrosomes (Fig. 6B,D). These findings demonstrate that the stepwise transitions in centriole engagement are crucial for proper centriole disengagement and subsequent conversion of daughter centrioles into functional centrosomes (Fig. 6E).

## Discussion

Centriole engagement is tightly maintained during interphase but rapidly disrupted after mitosis to ensure centrosome number control. However, the mechanisms underlying this precise temporal control remained elusive. Our study demonstrates that centriole engagement is tightly maintained through three redundant pathways: the cartwheel, torus, and PCM pathways (Fig. 6E). The cartwheel and torus pathways become sequentially dispensable before mitosis through two distinct steps of the daughter centriole maturation: centriole blooming and distancing, respectively. Release of the cartwheel and torus engagement pathways enables the rapid disruption of centriole engagement after mitosis, achieved through disassembly of the remaining PCM pathway. Thus, our findings show that the daughter centriole gradually matures throughout the cell cycle, ultimately achieving independence from its mother and becoming a functional centrosome.

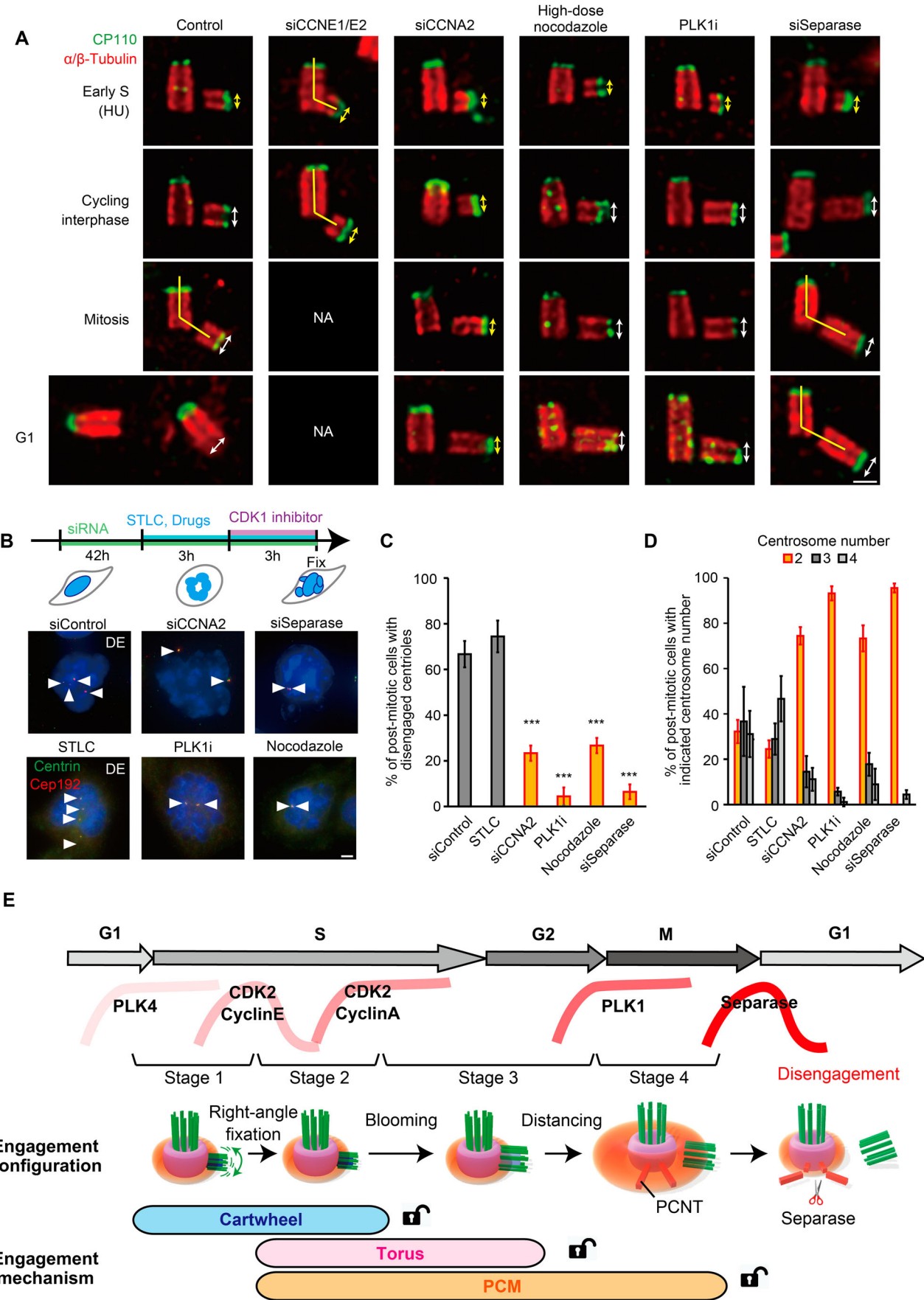

**Figure 6. The stepwise changes of the centriole engagement are critical for daughter centriole release and its centriole-to-centrosome conversion.**

(A) U-ExM images of centrioles in HeLa cells treated with indicated siRNAs (48 h) or drugs (3 h). The G1 images were taken after the treatment shown in (B). Scale bars: 200 nm. Yellow and white double-headed arrows indicate the width of thin and bloomed centrioles, respectively. (B) Representative immunofluorescence images of HeLa cells post-mitosis upon the treatments with indicated siRNAs or drugs. Scale bar: 5 μm. White arrowheads indicate the centrioles. (C) Quantification of the frequency of post-mitotic cells with disengaged centrioles in (B). Error bars represent the mean ± s.d. of three biological independent experiments. $P$ values were calculated by Dunnett's multiple comparisons test (siCCNA2, $P = 8.99E-05$; PLK1i, $P = 3.00E-08$; Nocodazole, $P = 9.91E-06$; siSeparase, $P = 2.82E-08$). \***$P < 0.001$. (D) Quantification of the frequency of the indicated centrosome numbers determined by the number of Cep192 foci in (B). Error bars represent the mean ± s.d. of three biological independent experiments. (E) Schematic model of the four stages of centriole engagement throughout the cell cycle. Each stage is characterized by a distinct mechanism and a unique configuration of the centriole engagement. The changes in the engagement are caused by the structural maturation of the daughter centriole. Source data are available online for this figure.

Before the establishment of stable orientation in Stage 2, the newly formed daughter centriole relies on the cartwheel for its initial engagement with the mother centriole (Stage 1). During Stage 1, the connection between the daughter and the mother centrioles is flexible, as evidenced by the variable engagement angles observed. Treatments that induced precocious disengagement readily disrupt the daughter centriole structure, highlighting its immaturity. This fragility is thought to play a role in limiting centriole numbers during duplication. If multiple potential duplication sites arise simultaneously, the final site is ultimately restricted to a single location (Ohta et al, 2014). The immaturity of the daughter centrioles likely allows for the elimination of any extra procentrioles, ensuring that only one daughter centriole is established. Upon reaching Stage 2, the daughter centriole establishes a stable connection with the mother centriole at a defined angle. Moreover, the Stage 2 daughter centriole is sufficiently stable to maintain its structure even after induced precocious disengagement. Interestingly, the transition from Stage 1 to Stage 2 is driven by CDK2-cyclinE complex activation, which coincides with the restriction point (Blagosklonny and Pardee, 2002). The restriction point is a critical cell cycle checkpoint beyond which the cell commits to division and cannot revert to a quiescent state (Blagosklonny and Pardee, 2002). Therefore, the restriction point serves as a point-of-no-return not only for the cell cycle progression but also for daughter centriole formation.

Following the restriction point, the centrioles establish orthogonally engagement. Orthogonal positioning of the daughter centriole relative to the mother has been observed in nearly all animals examined (Robbins et al, 1968; Marshall, 2009; Nabais et al, 2020). Our study provides the first evidence that the torus protein Cep63 is required for orthogonal engagement. In light of the role of the CDK2-cyclin E complex in orthogonal engagement formation in early S phase, we propose that this kinase complex may specifically target the torus proteins to initiate engagement. Interestingly, torus proteins, including Cep63, Cep57, and Cep57L1, lack homologs in nematodes and insects. Furthermore, several structural differences between centrioles of nematodes and insects compared to those in humans have been reported (Marshall, 2009; Nabais et al, 2020). In human centrioles, blooming involves diameter expansion and acquisition of triplet microtubules (Laporte et al, 2024). During blooming, the cartwheel's contribution to the engagement diminishes as the distance between the cartwheel and the daughter centriole's inner wall increases, before ultimately dissociating during mitosis. In contrast, in nematode and insect centrioles, which possess only singlet and doublet microtubules, respectively, the cartwheel remains associated even after mitosis (Marshall, 2009; Nabais et al, 2020). Therefore, the cartwheel might

maintain engagement throughout interphase in these organisms. Intriguingly, orthogonal engagement is also established in these species, suggesting that the orthogonal connection represents a robust and fundamental feature of centriole engagement, regardless of the specific mechanism involved.

Numerical abnormalities in centrosomes are known to contribute to cancer progression and malignancy (Levine et al, 2017). A primary cause of centrosome amplification is the disruption of centriole engagement (Wilhelm et al, 2019). In this study, we identified several molecules, including cyclin E, cyclin A, and PLK1, that regulate transitions of centriole engagement stages. Notably, overexpression of these molecules has been linked to centrosome amplification (Kawamura et al, 2004; Hanashiro et al, 2008; Gheghiani et al, 2021). These molecules are also components of the CA20 gene set, a recognized marker for both centrosome amplification and poor prognosis in cancer (Ogden et al, 2017). Based on these findings, we propose that centrosome amplification in cancer arises from centriole disengagement triggered by dysregulation of these molecules. Thus, targeting these molecules to stabilize centriole engagement could represent a promising therapeutic strategy to mitigate centrosome amplification and improve outcomes for cancer patients.

## Methods

**Reagents and tools table**

| Reagent/Resource | Reference or Source | Identifier or Catalog Number |
| --- | --- | --- |
| **Experimental models** | | |
| HeLa | ECACC | 93021013 |
| HeLa-Cas9 | Watanabe et al, 2019 | |
| HeLa-Cas9 Cep57KO | This study | |
| HeLa-Cas9 Cep57L1KO | This study | |
| HeLa-Cas9 Cep63KO | This study | |
| HeLa Tet3G-PLK1 T210D | Viol et al, 2020 | |
| **Recombinant DNA** | | |
| pCMV5-FLAG-Cep57 | Watanabe et al, 2019 | |
| pCMV5-FLAG-Cep57L1 | Ito et al, 2021 | |
| pCMV5-HA-Cep57 | Watanabe et al, 2019 | |

| Reagent/Resource | Reference or Source | Identifier or Catalog Number |
|---|---|---|
| pCMV5-HA-Cep63 | Ito et al, 2021 | |
| pTB701-FLAG-PCNT | Watanabe et al, 2019 | |
| **Antibodies** | | |
| Rabbit anti-ninein | Proteintech | 13007-1-AP |
| Rabbit anti-Cep57 | GeneTex | GTX115931 |
| Rabbit anti-Cep57L1 | Proteintech | 24957-1-AP |
| Rabbit anti-Cep63 | Proteintech | 16268-1-AP |
| Rabbit anti-PCNT | Abcam | ab4448 |
| Rabbit anti-Cep192 | Bethyl Laboratories | A302-324A |
| Rabbit anti-CP110 | Proteintech | 12780-1-AP |
| Rabbit anti-CENP-F | Abcam | ab108483 |
| Rabbit anti-Ac-tubulin | Abcam | Ab179484 |
| Rabbit anti-Cyclin E2 | Cell Signaling Tech. | 4132P |
| Rabbit anti-Phospho-Rb | Cell Signaling Tech. | 9308P |
| Mouse anti-Cep57 | Abcam | ab169301 |
| Mouse anti-PCNT | Abcam | ab28144 |
| Mouse anti-centrin | Merck | 20H5 |
| Mouse anti-SASS6 | Santa Cruz | Sc-81431 |
| Mouse anti-α-tubulin | Merck | DM1A |
| Mouse anti-Ac-tubulin | Sigma | T7451 |
| Mouse anti-CENPF | BD | 610768 |
| Mouse anti-Cyclin A2 | Cell Signaling Tech. | BF683 |
| Mouse anti-γ-tubulin | Merck | T5192 |
| Rat anti-PCNA | Abcam | ab252848 |
| Rat anti-centrin | BioLegend | 698602 |
| Guinea Pig anti-α-tubulin | ABCD Antibodies | AA345 |
| Guinea Pig anti-β-tubulin | ABCD Antibodies | AA344 |
| Goat anti-PCNT | Santa Cruz | Sc-28145 |
| Donkey anti-mouse IgG Alexa 488 | Invitrogen | A32766 |
| Donkey anti-rabbit IgG Alexa 555 | Invitrogen | A31572 |
| Donkey anti-rat IgG Alexa 647 | Invitrogen | A48272 |
| Camelid anti-rabbit IgG Abberior STAR 635P | Abonva | RAB00968 |
| Llama/alpaca anti-mouse IgG-X2 Abberior Star 635P | Progen | 1A23 |
| Goat anti-guinea pig IgG Alexa 555 | Invitrogen | A21435 |
| **Oligonucleotides and other sequence-based reagents** | | |
| siCep152 | Thermo Fisher | s225921 |
| siCep57 | Thermo Fisher | s18692 |
| siCep57L1 | Thermo Fisher | s226224 |
| siCep63 | Thermo Fisher | s37123 |

| Reagent/Resource | Reference or Source | Identifier or Catalog Number |
|---|---|---|
| siPCNT | Thermo Fisher | s10138 |
| siCCNA2 | Thermo Fisher | s2513 |
| siCCNE1 | Thermo Fisher | s2524 |
| siCCNE2 | Thermo Fisher | s17449 |
| siCep295 | Thermo Fisher | s229742 |
| siCPAP | Thermo Fisher | s31623 |
| siSeparase | Thermo Fisher | s18686 |
| Scrambled siRNA | Thermo Fisher | 4390843 |
| sgCep57 | This study | |
| sgCep57L1 | This study | |
| sgCep63 | This study | |
| **Chemicals, Enzymes and other reagents** | | |
| STLC | Sigma | 164739 |
| RO3306 | Sigma | SML0569 |
| BI2536 | AdooQ | A10134 |
| Hydroxyurea | Sigma | H8627 |
| Centrinone | MedChem Express | HY-18682 |
| CDK2 inhibitor III | Calbiochem | 238803 |
| Nocodazole | Wako | 140-08531 |
| Taxol | Tocris | 33069-62-4 |
| GSK461364 | MedChem Express | HY-50877 |
| **Software** | | |
| ZEN | Zeiss | |
| Fiji | Schindelin et al, 2012 | |
| **Other** | | |
| Tissue Culture Microplates (For Adherent Cells) 12 well | IWAKI | 3815-012 |
| 15 mm coverslip | Matsunami | C015001 |

## Methods and protocols

### Cell culture

HeLa cells were obtained from the European Collection of Authenticated Cell Cultures (ECACC). Cells were cultured in Dulbecco's modified Eagle's medium (DMEM) supplemented with 10% fetal bovine serum (FBS), 100 U/mL penicillin, and 100 μg/mL streptomycin at 37 °C in a humidified 5% $CO_2$ incubator. Transfection of siRNA or DNA constructs into HeLa was conducted using Lipofectamine RNAiMAX (Life Technologies) or Lipofectamine 2000 (Life Technologies), respectively. Unless otherwise noted, the transfected cells were analyzed 48 h after transfection with siRNA and 24 h after transfection with DNA constructs.

### Generation of knockout cell lines

To generate knockout cell lines, we utilized HeLa cells stably expressing Cas9 (HeLa-Cas9), which were previously established

using a lentiviral system (Watanabe et al, 2019). Single guide RNAs (sgRNAs) were designed to target exons in the N-terminal region that are common to all isoforms of the target gene. The sgRNAs were synthesized by in vitro transcription from DNA oligonucleotide templates using the HiScribe T7 Transcription Kit (New England Biolabs). The transcribed products were subsequently purified using the RNA Clean & Concentrator (ZYMO RESEARCH). Purified sgRNAs were introduced into HeLa-Cas9 cells by transfection using Lipofectamine RNAiMAX (Life Technologies). Following transfection, single-cell colonies were isolated by limiting dilution. The successful generation of the knockout cell lines was confirmed by the absence of an immunofluorescence signal in microscopy and by sequencing of the targeted genomic region.

### Immunofluorescence staining (IF)

Cells cultured on 15 mm diameter, 0.12–0.17-mm-thick round coverslips were fixed by immersion in methanol at $-20\,^{\circ}\text{C}$ for 7 min. The fixed cells were then washed three times with PBS and blocked for 30 min at room temperature in a PBS solution containing 1% BSA and 0.05% Triton X-100 (blocking buffer). Subsequently, the cells were incubated with primary antibodies diluted in blocking buffer for 2 h at room temperature. After three washes with PBS, the cells were incubated with secondary antibodies in blocking buffer for 2 h at room temperature. Finally, the cells were stained with Hoechst in PBS and mounted on glass slides using 90% glycerol.

For STED microscopy imaging of centrioles, cells were first placed at $4\,^{\circ}\text{C}$ for 30 min to depolymerize cytoplasmic microtubules. The cells were then permeabilized by incubation with CSK buffer (25 mM HEPES pH 7.4, 50 mM NaCl, 1 mM EDTA, 3 mM $MgCl_2$, 300 mM Sucrose, 0.5% Triton X-100) for 5 min before methanol fixation. Following fixation, the cells were processed identically as described above.

### Microscopy

Fluorescence microscopy imaging and phenotype counting were performed using an Axioplan2 fluorescence microscope (Carl Zeiss) with 63x/1.4 NA plan-APOCHROMAT objectives. STED microscopy images were acquired using a Leica TCS SP8 STED 3X system with a Leica HC PLAPO 100x/1.40 oil STED WHITE objective, and a 775 nm gated STED laser.

### Ultrastructure expansion microscopy (U-ExM)

The U-ExM protocol was performed as described previously (Gambarotto et al, 2021). Cells cultured on 12 mm diameter, 0.12–0.17-mm-thick round coverslips were incubated in 2% acrylamide + 1.4% formaldehyde diluted in PBS for 3–12 h. The coverslips were then placed on 35 μL of monomer solution (19% sodium acrylate, 0.1% bis-acrylamide, and 10% acrylamide) with 0.5% TEMED and 0.5% APS for 5 min at $4\,^{\circ}\text{C}$ and then for 1 h at $37\,^{\circ}\text{C}$. The gels were then incubated in the denaturation buffer (200 mM SDS, 200 mM NaCl, 50 mM Tris pH 9) and boiled at $95\,^{\circ}\text{C}$ for 90 min. Next, they were transferred into water at room temperature. After expansion, the gels were cut into quarters. The gels were then incubated with primary antibodies diluted in a blocking buffer for more than 3 h at $37\,^{\circ}\text{C}$. After three washes with PBS, the cells were incubated with secondary antibodies in the blocking buffer for 2 h at $37\,^{\circ}\text{C}$. The expanded samples were mounted onto glass-bottom dishes with minimal residual water and

imaged using the Leica TCS SP8 HSR confocal system. Images were captured with a Leica HCX PL APO ×63/1.4 oil CS2 objective lens and excitation wavelengths of 405, 488, and 561 nm.

### Quantification

For the quantification of centriole width (Figs. 1D, 3F, EV1D–F, and EV2J), length (Fig. EV1D–F), the position of the edge of the torus and PCM ring (Figs. 4B and 5M), the distance between mother and daughter centrioles (Figs. 5F and EV4F), the length of non-acetylated microtubules at the daughter's proximal end (Fig. 5G), and cartwheel length (Fig. 5I), we measured the distance between two endpoints. Each endpoint was defined as the position where the signal intensity dropped to half of the maximum. The precise location of this half-intensity point was calculated by interpolating between the positions of the two pixels closest to this half-maximal intensity value.

In Fig. EV4H, we assessed whether the daughter/new mother centrioles retained γ-TuRC at their proximal ends based on γ-tubulin intensity. For γ-tubulin intensity quantification, three regions were measured on each centriole. First, two proximal ends were identified and enclosed in circles with an 8-pixel (corresponding to 240 nm) diameter to quantify the intensity at these sites. Next, the proximal lumen side was measured by placing a 12-pixel (360 nm) diameter circle in a position that avoids overlap with the lumen pool γ-tubulin. Finally, a 12-pixel diameter circle was placed over an area devoid of any signal to serve as the background measurement. Mean intensity values were obtained for each region. A centriole was classified as γ-TuRC positive if the intensity at both proximal ends, relative to the lumen side, exceeded a set threshold (1.8-fold).

## Data availability

This study includes no data deposited in external repositories.

The source data of this paper are collected in the following database record: biostudies:S-SCDT-10_1038-S44318-024-00350-8.

## Peer review information

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

## Acknowledgements

We thank Mariya Genova for the proofreading of the manuscript, Paul Guichard and Virginie Hamel for providing guidance on U-ExM, and the Kitagawa lab members for technical support and helpful discussions. This work was supported by JSPS KAKENHI grants (Grant numbers: 18K06246, 19H05651, 20K15987, 20K22701, 21H02623, 21J22462, 22H02629, 22K20624, 22KJ0633, 22KJ0687, 23K14176, 23KJ0800, 23H02627, 24K02174, 24H02284) from the Ministry of Education, Science, Sports and Culture of Japan, the PRESTO program (JPMJPR21EC) and the CREST program (JPMJCR22E1) of the Japan Science and Technology Agency, Takeda Science Foundation, The Uehara Memorial Foundation, The Research Foundation for Pharmaceutical Sciences, Koyanagi Zaidan, The Kanae Foundation for the Promotion of Medical Science, Kato Memorial Bioscience Foundation, Naito Foundation, Heiwa Nakajima Foundation, Sumitomo Foundation, Inamori foundation, Astellas Foundation for research on metabolic disorders, and Tokyo Foundation for Pharmaceutical Sciences.

## Author contributions

**Kei K Ito**: Conceptualization; Data curation; Formal analysis; Funding acquisition; Validation; Investigation; Visualization; Methodology; Writing—original draft; Project administration; Writing—review and editing. **Kasuga Takumi**: Data curation; Formal analysis; Visualization; Methodology; Writing—review and editing. **Kyohei Matsuhashi**: Formal analysis; Methodology; Writing—review and editing. **Hirokazu Sakamoto**: Methodology; Writing—review and editing. **Kaho Nagai**: Data curation; Writing—review and editing. **Masamitsu Fukuyama**: Methodology; Writing—review and editing. **Shohei Yamamoto**: Methodology; Writing—review and editing. **Takumi Chinen**: Funding acquisition; Methodology; Writing—review and editing. **Shoji Hata**: Conceptualization; Formal analysis; Supervision; Funding acquisition; Investigation; Methodology; Writing—original draft; Project administration; Writing—review and editing. **Daiju Kitagawa**: Conceptualization; Formal analysis; Supervision; Funding acquisition; Investigation; Methodology; Writing—original draft; Project administration; Writing—review and editing.

Source data underlying figure panels in this paper may have individual authorship assigned. Where available, figure panel/source data authorship is listed in the following database record: biostudies:S-SCDT-10_1038-S44318-024-00350-8.

## Disclosure and competing interests statement

The authors declare no competing interests.

# Expanded View Figures

**Figure EV1.   Effect of HU and CDK2 inhibitor treatments on centriole and cell cycle.**

(**A**) Representative immunofluorescence images of HeLa cells treated with HU or HU + CDK2 inhibitor. Scale bar: 5 µm. (**B**) Quantification of the p-Rb (Ser807/811) signal intensity in the nucleus from (**A**). Boxplot was employed to illustrate the data distribution. The central line represents the median, the box spans the interquartile range (IQR) from the 25th to the 75th percentile, and the whiskers extend to the smallest and largest data points within 1.5 times the IQR. *P* values were calculated by two-tailed unpaired Student's t-test ($P = 1.57E{-}08$). ***$P < 0.001$. (**C**) Quantification of the DNA content in HeLa cells treated as indicated, using flow cytometric analysis. (**D–F**) Scatter plot quantification of the length and the width of the daughter centrioles from Fig. 1. (**G**) Representative immunofluorescence images of HeLa cells in early S phase, late S phase, G2 phase, and mitosis. Scale bars: 5 µm, 1 µm. (**H**) Quantification of the frequency of cells with disengaged centrioles in G2 phase in (**G**). Error bars represent the mean ± s.d. of three biological independent experiments. *P* value was calculated by Dunnett's multiple comparisons test ($P = 1.24E{-}07$). ***$P < 0.001$. (**I**) Quantification of the frequency of cells with disengaged centrioles in mitosis in (**G**). Error bars represent the mean ± s.d. of three biological independent experiments. *P* value was calculated by Dunnett's multiple comparisons test ($P = 8.06E{-}10$). ***$P < 0.001$. Source data are available online for this figure.

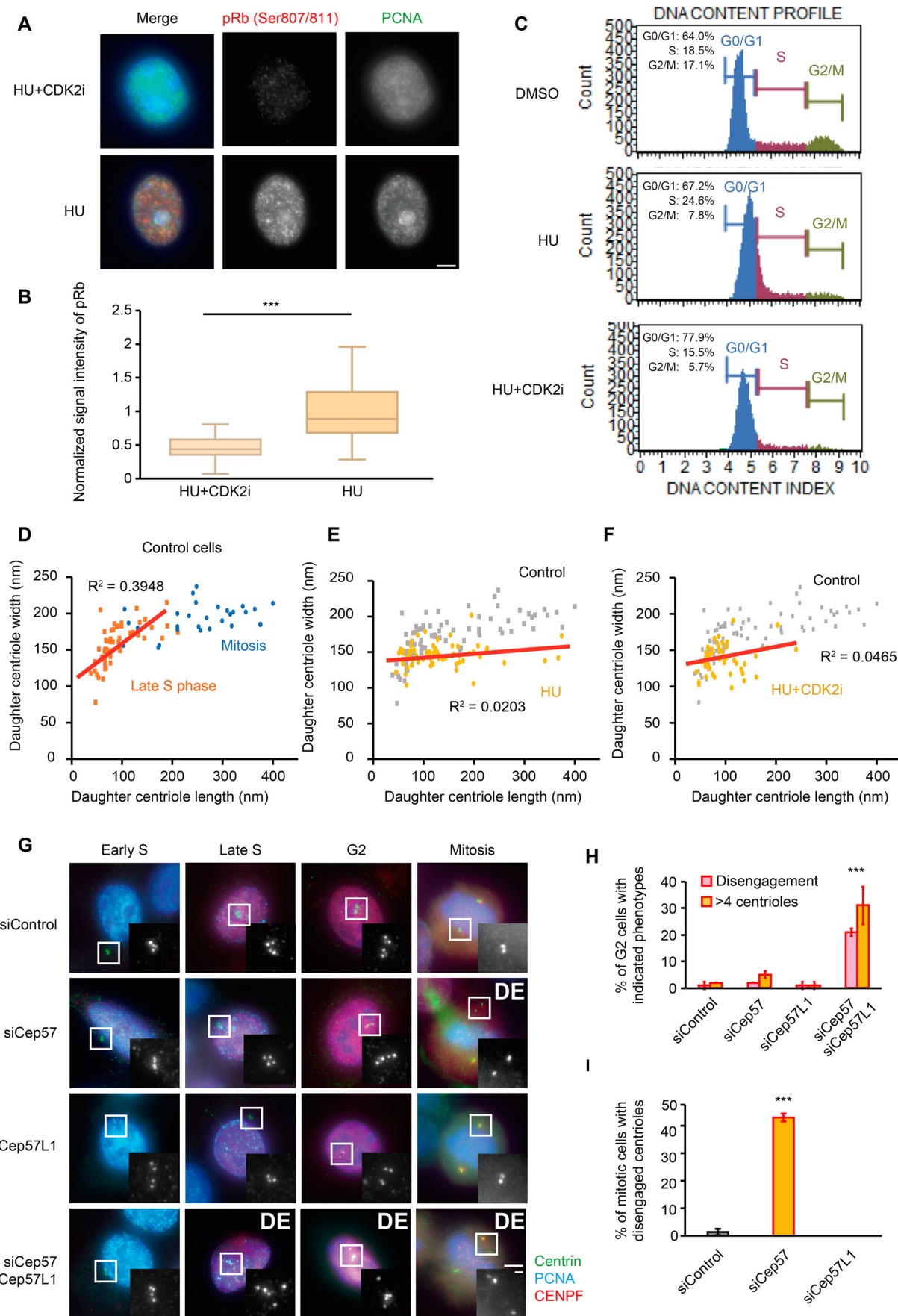

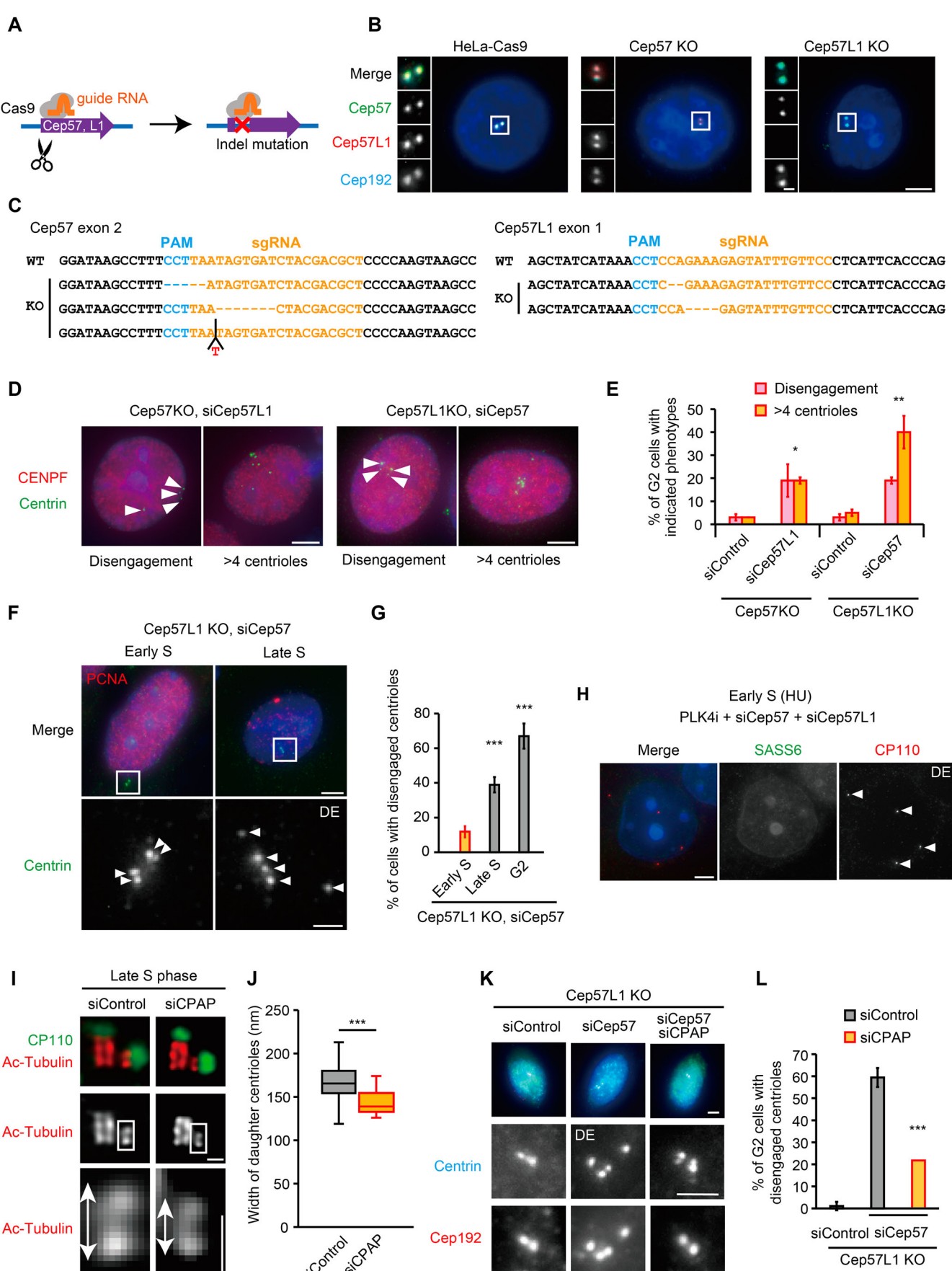

◀ **Figure EV2. Confirmation of the centriole engagement by Cep57 and Cep57L1 in interphase using knockout cell lines.**

(A) Schematic illustration for generating knockout (KO) cells using CRISPR-Cas9. (B) Immunofluorescence images of HeLa-Cas9, Cep57KO, and Cep57L1KO cells. Scale bars: 5 μm, 1 μm. (C) DNA sequences surrounding the CRISPR-targeted regions in the exons of the Cep57 and Cep57L1 genes. (D) Representative immunofluorescence images of HeLa-Cep57KO cells transfected with siCep57L1, and Cep57L1KO cells transfected with siCep57 in G2 phase. Scale bars: 5 μm. White arrowheads indicate the centrioles. (E) Quantification of the frequency of cells possessing four disengaged centrioles and more than four centrioles in G2 phase in (D). Error bars represent the mean ± s.d. of three biological independent experiments. *P* value was calculated by two-tailed unpaired Welch's t-test with Bonferroni correction (Cep57KO, $P = 0.0230$; Cep57L1KO, $P = 0.00146$). *$P < 0.05$; **$P < 0.01$. (F) Representative immunofluorescence images of HeLa-Cep57L1KO cells transfected with siCep57 in early and late S phase. Scale bars: 5 μm, 1 μm. White arrowheads indicate the centrioles. (G) Quantification of the frequency of the cells with disengaged centrioles in early S, late S, and G2 phase in (F). Error bars represent the mean ± s.d. of three biological independent experiments. *P* values were calculated by Dunnett's multiple comparisons test (Late S, $P = 0.000192$; G2, $P = 1.70E-06$). ***$P < 0.001$. (H) Representative immunofluorescence images of HeLa cells subjected to a cartwheel removal assay. HeLa cells were treated with HU and CDK2 inhibitor III (CDK2i) for 24 h, followed by treatment with a PLK4 inhibitor for another 24 h. Scale bar: 5 μm. White arrowheads indicate the centrioles. (I) Representative STED microscopy images of HeLa cells transfected with siControl or siCPAP. Double-headed arrows indicate the width of centrioles. Scale bars: 200 nm. (J) Quantification of the width of the daughter centrioles in (I). $n = 40$ centrioles pooled from 3 independent experiments. *P* value was calculated by two-tailed unpaired Welch's t-test ($P = 5.54E-07$). ***$P < 0.001$. (K) Representative immunofluorescence images of HeLa Cep57L1 KO cells transfected with siControl, siCep57, or siCep57 + siCPAP. Scale bars: 5 μm, 1 μm. (L) Quantification of the frequency of cells with disengaged centrioles in (K). Error bars represent the mean ± s.d. of three biological independent experiments. *P* value was calculated by two-tailed unpaired Welch's t-test ($P = 0.000419$). ***$P < 0.001$. Source data are available online for this figure.

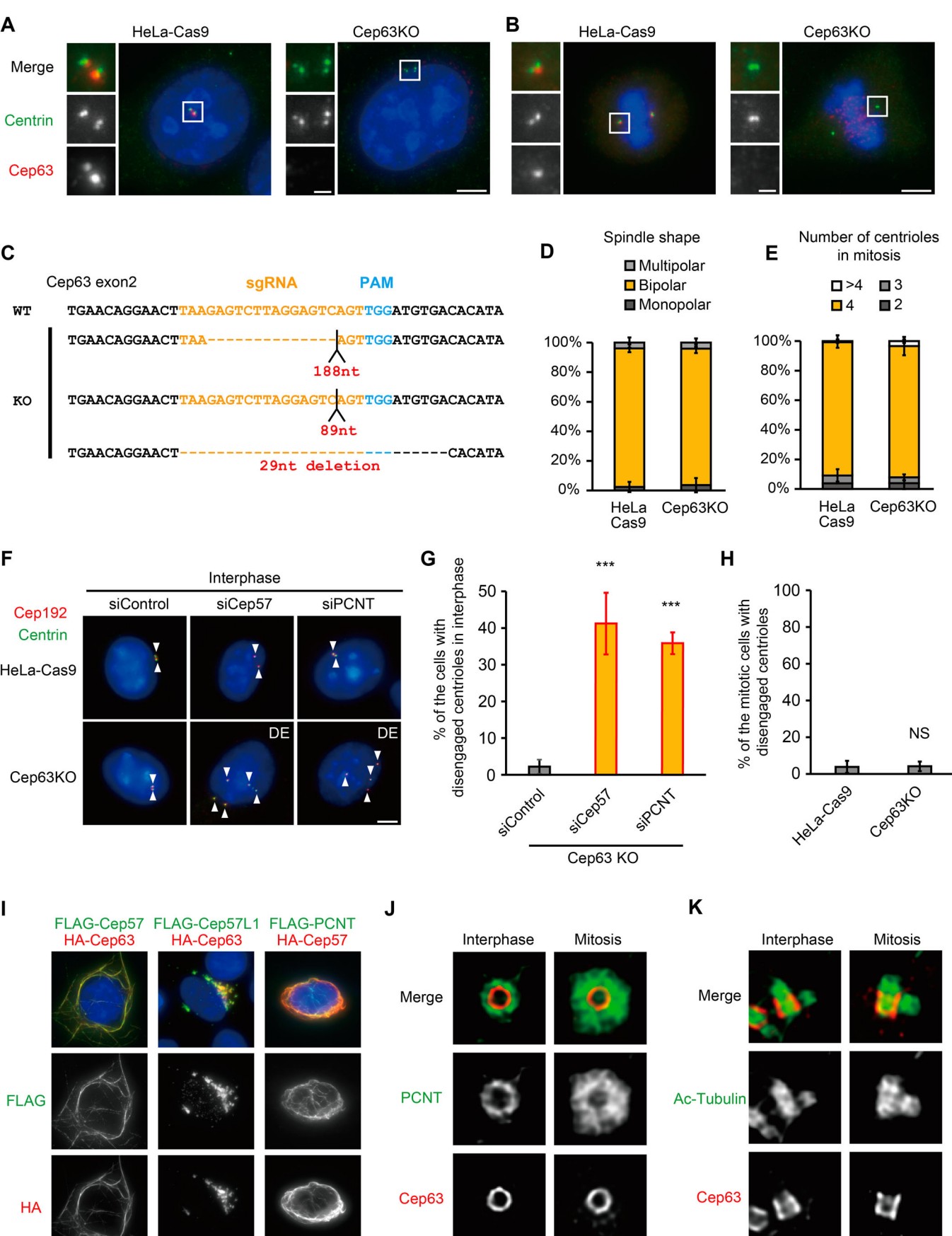

◀ **Figure EV3. Cep63 is involved in centriole engagement.**

(A) Immunofluorescence images of HeLa-Cas9 and Cep63KO cells in interphase. Scale bars: 5 µm, 1 µm. (B) Immunofluorescence images of HeLa-Cas9 and Cep63KO cells in mitosis. Scale bars: 5 µm, 1 µm. (C) DNA sequences surrounding the CRISPR-targeted regions in the exons of the *Cep63* gene. (D) Quantification of the frequency of mitotic cells with the indicated spindle shape. Error bars represent the mean ± s.d. of three biological independent experiments. (E) Quantification of the frequency of mitotic cells with the indicated number of centrioles in (B). Error bars represent the mean ± s.d. of three biological independent experiments. (F) Representative immunofluorescence images of HeLa-Cas9 and Cep63KO cells transfected with siControl, siCep57 or siPCNT in interphase. Scale bar: 5 µm. White arrowheads indicate the centrioles. (G) Quantification of the frequency of cells with disengaged centrioles in (F). Error bars represent the mean ± s.d. of three biological independent experiments. *P* values were calculated by Dunnett's multiple comparisons test (siCep57: *P* = 0.000234, siPCNT: *P* = 0.000496). \*\*\**P* < 0.001. (H) Quantification of the frequency of mitotic cells with disengaged centrioles. Error bars represent the mean ± s.d. of three biological independent experiments. *P* value was calculated by two-tailed unpaired Welch's t-test (*P* = 0.912). NS, not significant. (I) Immunofluorescence images of HeLa cells overexpressing Cep57, Cep57L1, Cep63, or PCNT after plasmid transfection. Scale bar: 5 µm. (J) STED microscopy image showing the top view of a mother centriole in G2 and mitosis. Scale bar: 200 nm. (K) STED microscopy image showing the side view of a mother centriole in G2 and mitosis. Scale bar: 200 nm. Source data are available online for this figure.

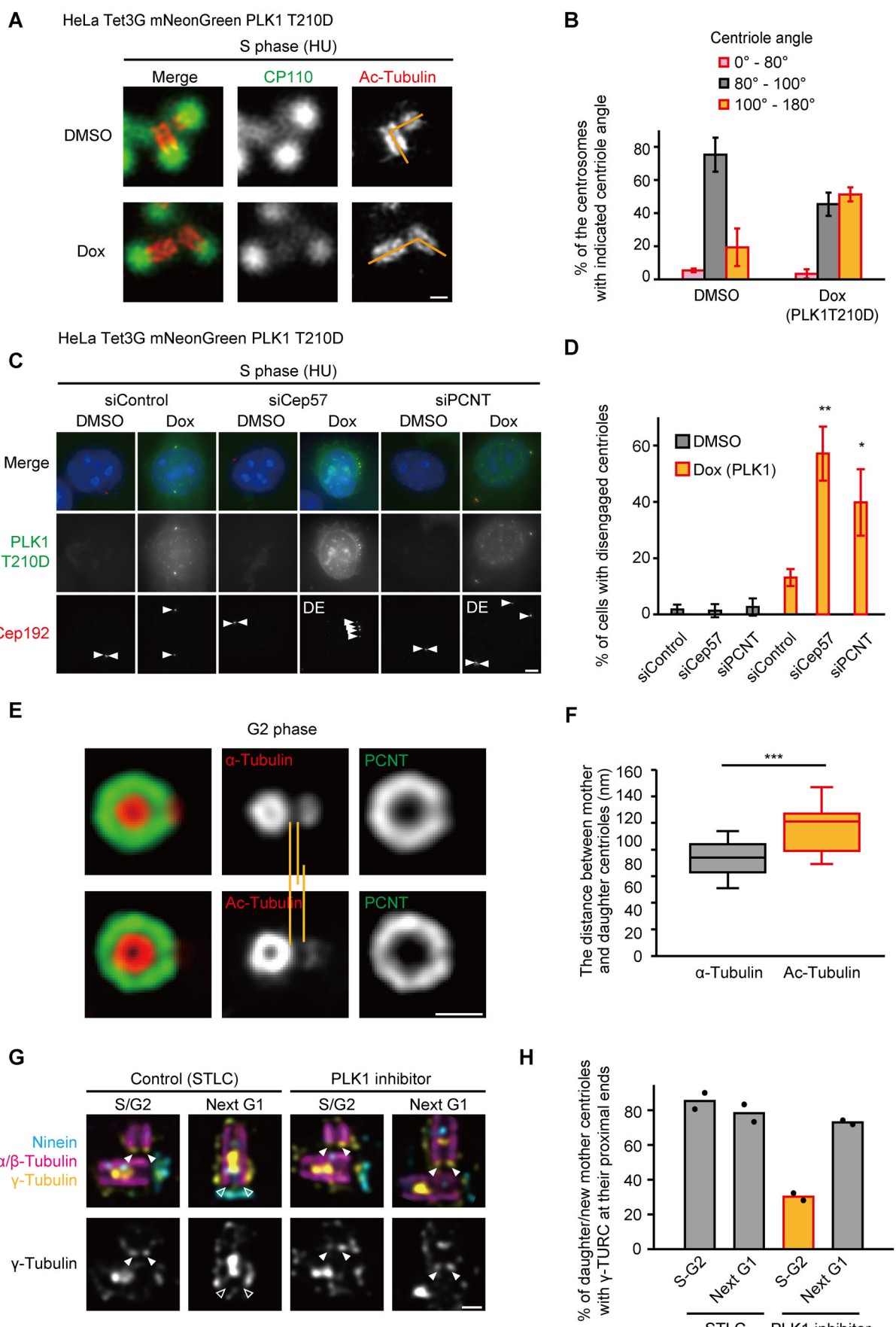

**Figure EV4.   PLK1 overexpression changes the configuration and the mechanism of centriole engagement.**

(A) Representative STED microscopy images of HeLa-tet3G-PLK1T210D cells treated with DMSO or Doxycycline in HU-arrested S phase. Scale bar: 200 nm.
(B) Quantification of the frequency of cells with indicated centriole angle in (A). Error bars represent the mean ± s.d. of three biological independent experiments.
(C) Representative immunofluorescence images of HeLa-tet3G-PLK1T210D cells treated with DMSO or Doxycycline in HU-arrested S phase. Scale bar: 5 μm. White arrowheads indicate the centrioles. (D) Quantification of the frequency of cells with disengaged centrioles in (C). Error bars represent the mean ± s.d. of three biological independent experiments. $P$ values were calculated by Dunnett's multiple comparisons test (HU + Dox + siCep57: $P = 0.00490$, siPCNT: $P = 0.0447$). *$P < 0.05$; **$P < 0.01$. (E) Representative STED microscopy images of centrioles in G2 phase immunostained with anti-PCNT and anti-α-tubulin or anti-acetylated tubulin antibodies. Scale bar: 200 nm. (F) Quantification of the distance between the mother centriole wall and the proximal end of the daughter centriole in (E). Boxplot was employed to illustrate the data distribution. The central line represents the median, the box spans the interquartile range (IQR) from the 25th to the 75th percentile, and the whiskers extend to the smallest and largest data points within 1.5 times the IQR. $P$ value was calculated by two-tailed unpaired Student's t-test ($P = 0.000329$). ***$P < 0.001$.
(G) Representative U-ExM images of daughter and new mother centrioles in HeLa cells treated with STLC or the PLK1 inhibitor BI2536. New mother centrioles were obtained from cells forced into G1 by treatment with the prometaphase-arresting drugs STLC or BI2536 for 3 h, followed by the addition of the CDK1 inhibitor RO3306 (10 μM) for 3 h. New mother centrioles are identified by the absence of ODF2 signal at the distal part. Filled and open arrowheads indicate the proximal end of daughter/new mother centrioles with and without γ-tubulin signals, respectively. Scale bar: 200 nm. (H) Quantification of the frequency of the daughter or new mother centrioles with γ-tubulin signal at its proximal end in (G). Two biologically independent experiments, 30 cells each. Details of the quantification method are provided in the Methods section. Source data are available online for this figure.

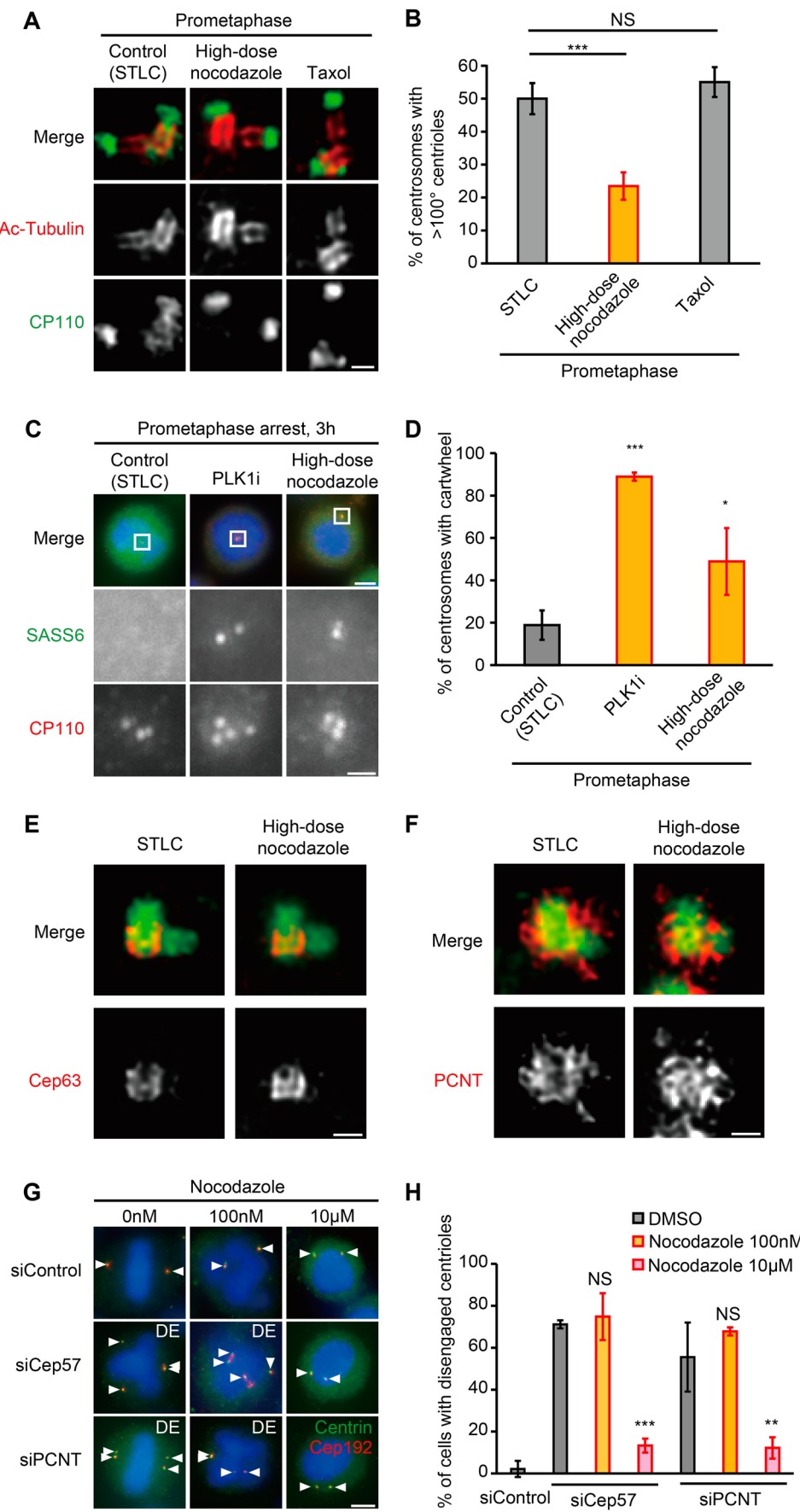

◄ **Figure EV5. The proximal end of the daughter centriole is depolymerized in mitosis.**

(A) Representative STED microscopy images of centrioles in mitosis treated with DMSO, high-dose nocodazole (10 µM), or taxol for 3 h. Scale bar: 200 nm.
(B) Quantification of the frequency of the centrosomes, in which the centrioles are connected at an obtuse angle in (A). Error bars represent the mean ± s.d. of three biological independent experiments. *P* values were calculated by Dunnett's multiple comparisons test (Nocodazole: $P = 0.000458$, Taxol: $P = 0.291$). NS, not significant; ***$P < 0.001$. (C) Representative immunofluorescence images of HeLa cells in mitosis treated with STLC, PLK1 inhibitor, or high-dose nocodazole for 3 h. Scale bar: 5 µm, 1 µm. (D) Quantification of the frequency of cells with cartwheel-retaining centrosomes in (C). Error bars represent the mean ± s.d. of three biological independent experiments. *P* values were calculated by Dunnett's multiple comparisons test (PLK1i: $P = 0.00029$, Nocodazole: $P = 0.0185$). *$P < 0.05$; ***$P < 0.001$. (E, F) Representative STED microscopy images of centrioles in HeLa cells. Scale bars: 200 nm. (G) Representative immunofluorescence images of HeLa cells in mitosis transfected with siControl, siCep57, or siPCNT, under treatment with different doses of nocodazole. Scale bar: 5 µm. White arrowheads indicate the centrioles. (H) Quantification of the frequency of cells with disengaged centrioles in mitosis in (G). Error bars represent the mean ± s.d. of three biological independent experiments. *P* values were calculated by Dunnett's multiple comparisons test (siCep57 100 nM nocodazole: $P = 0.739$, 10 µM nocodazole: $P = 0.000163$; siPCNT 100 nM nocodazole: $P = 0.299$, 10 µM nocodazole: $P = 0.00297$). NS, not significant; **$P < 0.01$; ***$P < 0.001$. Source data are available online for this figure.

