## [Peer Review File · The EMBO Journal]

Multimodal mechanisms of human centriole engagement and disengagement

Kei Ito, Kasuga Takumi, Kyohei Matsuhashi, Hirokazu Sakamoto, Kaho Nagai, Masamitsu Fukuyama, Shohei Yamamoto, Takumi Chinen, Shoji Hata, and Daiju Kitagawa

Corresponding author(s): Daiju Kitagawa (dkitagawa@mol.f.u-tokyo.ac.jp) , Shoji Hata (s.hata@mol.f.u-tokyo.ac.jp)

Review Timeline:

Transfer from Review Commons:	15th Oct 24
Editorial Decision:	29th Nov 24
Revision Received:	11th Dec 24
Accepted:	12th Dec 24

Review
COMMONS

Editor: Hartmut Vodermaier

Transaction Report: This manuscript was transferred to The EMBO JOURNAL following peer review at Review Commons.

Review #1

1. Evidence, reproducibility and clarity:

Evidence, reproducibility and clarity (Required)

The article by Ito and colleagues focuses on the mechanisms that govern centriole disengagement. This process is crucial for the function of the centrosome. In G1, the centrosome consists of two mature disengaged centrioles, each of which will initiate a procentriole at the beginning of S-phase. Throughout the S-phase, G2, and up to mitosis, the procentriole will elongate until it forms a new daughter centriole, which will disengage during mitosis. If this process is altered, the function of the centrosome is compromised and will lead to an abnormal number of centrioles, centrosomes, and mitotic errors.

The mechanisms and molecular players of this engagement are not well known. The same group and others had previously shown that Cep57, Cep57L1, and PCNT were involved in this process. Here, the authors propose a comprehensive study on the disengagement processes and identify several distinct stages during centriole growth: engagement associated with the cartwheel, the torus, and the PCM. The authors also identify factors that regulate the orthogonality of the procentriole, a characteristic that had not been deeply studied before. The description of these stages, coupled with the cell cycles and structural changes of the centriole, provides a better understanding of how this crucial process works.

This work is of high importance for the field. The authors have done meticulous work with a large number of diverse methods, including very well-executed cellular biology approaches coupled with super-resolution microscopy (STED, U-ExM). However, I find that the article is somewhat complex, and some interpretations/conclusions do not always seem correct, or some are overstated (My comments are not ranked in order of importance):

- In the introduction, the authors say that there are two centrosomes in G1. Shouldn't there be 1 centrosome with two centrioles instead?
- I am confused between mother centriole, daughter centriole, and procentriole. I think it would be easier to understand if the term procentriole is used until the end of mitosis, giving rise to a daughter centriole in G1, disengaged from the mother. In fact, the disengagement should be the transition from procentriole to daughter centriole.
- The authors link the early steps of procentriole formation and disengagement with the bloom, a recently described structural change of the centriole. In the cited article, the bloom is initiated from the formation of centriole microtubules, whereas here the authors talk about the bloom as a later stage, end of S, which is between stage 2 and stage 3 of the disengagement process described here. It seems to me that the authors should review their analysis and model considering that the bloom is initiated at the beginning of procentriole formation. The bloom step likely appears during stage 1 during orthogonal adjustment.
- The authors correlate centriole diameter and the cell cycle (for example in figure 1d). How do the authors know they are in late S and not in G2? If the cells are synchronized, I did not find the information in the materials and methods section. To relate the cell cycle to procentriole growth, it would be good to correlate diameter with the length of centrioles for a better idea of the

procentriole growth stage. This type of analysis could also show that centrioles with HU or HU+Cdk2i treatment have a length that corresponds not to an S phase but rather to a G2 (based on the figures. In figure 3i, the procentriole's length appears correct but in figure 1b procentrioles are very long, like in G2).

- In the same idea of relating the 4 described stages with the cell cycle, stage 2 is probably in late S phase and not early. Therefore, cell synchronization should be done (by mitotic shake-off because treatments with inhibitors may affect centriole length). This analysis would help refine the model and could change some conclusions.

- Regarding the experiment in figure 3e with STED, the authors measured the width using Ac tubulin antibody. Acetylated tubulin staining evolves during growth so if the centrioles are not at the same growth stage, the values can't be comparable. Here procentriole seems to be at different stages. An analysis with tubulin staining would be more accurate. Moreover, I am not sure I understand, is it the width or length that is measured? The line indicates the length.

- Regarding acetylation not covering the proximal part of the procentriole, this result is interesting and suggests a proximal disassembly of the microtubule triplets. Nevertheless, this is staining with two antibodies that do not detect the same epitopes, so the difference could also come from the staining and accessibility. It can also be that tubulin acetylation is gradual and covers the proximal later in the cell cycle. If the tubulin is trimmed on the proximal side, one would imagine that other proximal markers would disappear in G2. For example, is the gamma-TuRC capping the A-microtubule removed? Moreover, the change in angle observed with nocodazole may also be a distancing effect due to the relaxation of the PCM.

- The authors report that CPAP depletion suppresses precocious centriole disengagement in mitosis under depletion of Cep57 or PCNT. CPAP has also been shown to be important for the formation of microtubule triplets and CPAP depletion leads to incomplete centrioles that fragment (DOI: 10.1083/jcb.202108018). Therefore, I'm not sure to understand the link the authors propose with the loss of the proximal region.

- The result with Cep63 is very interesting. In the images, it seems that the distance between the wall of the mother centriole and the procentriole is also reduced. Have you quantified this? If the distance is smaller, does this suggest that the procentriole is in phase 1? Does the angle remain obtuse throughout growth?

2. Significance:

Significance (Required)

The article by Ito et al. is very interesting and focuses on the disengagement mechanisms of human centriole. The authors have dissected the process and revealed new, previously unknown steps that follow one another throughout the cell cycle. They have also linked centriole engagement with its orthogonality, a feature of the centriole that is still poorly understood. This article is therefore important for the field. Nevertheless, some analyses do not appear to be fully synchronized with the cell cycle, which may lead to confusion and possibly affect conclusions about the timing of the various steps described.

The major issue concerns the description of timings, which seems to differ from previous observations. An analysis in line with the cell stage would be important here.

3. How much time do you estimate the authors will need to complete the suggested revisions:

Estimated time to Complete Revisions (Required)

(Decision Recommendation)

Between 3 and 6 months

Yes

Review #2

1. Evidence, reproducibility and clarity:

Evidence, reproducibility and clarity (Required)

****Summary****

This study explores the events of centriole engagement and disengagement. Centriole engagement dynamics are important because engagement ensures that centriole overduplication does not occur, but cells must also be prepared for disengagement during mitosis to enable centriole duplication during G1/S of the following cell cycle. This study characterizes four stages for centriole engagement dynamics: 1) Stage 1: at G1/S boundary with flexible connection of the daughter centriole to the mother centriole wall becoming orthogonal, 2) Stage 2: promotes blooming of centriole width that, based on prior studies, suggesting centrioles of singlet MTs gaining triplet MTs, 3) Stage 3: further distancing by loss of the proximal centriole during mitosis. Finally, disengagement occurs during separase cleavage in mitosis. The cartwheel, torus, and PCM and associated molecules are all required for unique and overlapping phases of the engagement. While some of these stages and events are reported in prior work, this study provides a comprehensive view of centriole engagement. Overall, the study is thorough and provides important insights for how centriole engagement/disengagement occurs. In places, I struggled with the complexity of the manuscript as currently written, and provide details for how I believe the manuscript can be improved below.

****Major****

Generally, the manuscript is written for the expert and could be improved by simplifying the text and findings and clearly explaining the various perturbations and treatments provided. It was easy to get

bogged down in the details of this manuscript. Clearer/tighter writing would improve the impact of the paper. Reader is asked to wade through a lot of jargon and detail of the experiments to follow. There is a lot of "we did..", I wonder if this can be told as a more simple story to really get things going.

Perhaps having an outside reader help to shorten the text to more concise statements would address this issue.

If CycE/CDK2 is promoting stabilization of the new daughter centriole into an orthogonal position, what is promoting new centriole assembly. Is Plk4 only required for assembly in the canonical cell cycle, while CycE/CDK2 is not required for new centriole assembly? This delineation of roles for new centrioles forming independent of the CDK2 and cyclin E and/or A activity seems important to describe.

Perhaps most surprising is the loss of the proximal end of the centriole. This is a non-acetylated tubulin domain. Do centrioles get shorter? I think this needs to be flushed out more by costaining MTs generally and then the Ac Tub specifically. The proposal that the proximal region of the centriole and CW is disassembled is interesting, however I don't think that this is well established from these data. First, I do not see support for this from prior studies, like Laporte et al 2024, who co-localized Actubulin and tubulin. From these studies, it appears that Actubulin is at the proximal region early in assembly and is maintained there. Can the authors study both markers in the context of their perturbations to establish this? Are the gamma-tubulin capped minus ends of the A-MTs lost? What is the predicted length for the CW and does that correlate with the 30nm of MTs that is proposed to be lost? Do the authors observe the centriole to shorten at this step of separation? Beyond the above comments, I think a clear Discussion of what might promote this and how it could be controlled is necessary.

Does the high nocodazole treatment affect cell cycle progression, and could that produce the described result of no separation/distanced centrioles? Could high drug treatment arrest the cells earlier and this blocks "distancing"? Please address.

It was not clear to me that the Cdk2i+HU resulted in an earlier arrest. Was this timing of the cell cycle confirmed? Are there markers that the authors could use beyond just the statement that orthogonal positioning is now changed? Or can you show this by FACS analysis of DNA replication? Or centrin maturation?

Methods for measurements: The methods for measuring and calibrating to the ExM samples needs to be explained in the Materials and Methods section. For one example, how was centriole width defined? I assume that this was based on a full width half maximum of a certain criterion. This should be defined.

Figure 2C,D - 30% of the cells exhibited precocious disengagement. What happened to the remaining 70%?

Figure 2F - Why did the authors focus on quantifying 80-100deg and not report the SD and coefficient of variation? It seems like this would be more accurate and a stronger description of the variance?

Despite identifying cell cycle (cycE/CDK2) and molecular factors (Cep63) responsible, it would be interesting if the Discussion focused more on how the unique events of Stage 1 to Stage 2 occurs. What is a mechanistic model for this? Similarly, why would blooming stabilize the links? I think this

should at least be a discussion point. I found this same theme for the codependences between Stage 3 factors as well. It was not clear why these factors would be codependent. I think this can also be addressed in the discussion.

****Minor****

Define the torus and PCM for general readers.

To be clearer, I would suggest that "after cell division" is stated at "during G1" or whatever the authors intend. I found this statement could be more precise. Same for "end of mitosis."

There are some minor grammatical mistakes that should be addressed.

There are a number of places where the references should be expanded to provide context for these results relative to the literature in the field. An example is the CW engagement role. Guichard et al 2010 EMBO have structural data showing such an attachment.

How does centrinone treatment remove the CW? This was not well explained.

I was unclear as to why in some places "Cdk2" was used and others "CDK2" was used. Please keep consistent.

I was not clear how Figure 2E showing "CDK2 is activated by forming complexes....". Please clarify.

2. Significance:

Significance (Required)

Overall, this is an interesting paper that identifies important findings associated with the field of centriole engagement and disengagement dynamics. This will be important for the centriole field but may be less interesting to a general audience. The work appears to be associated with prior studies from the Loncerak lab on Plk1 control of centriole separation during mitosis, the Rhee lab, and others. Some of this work relates to recent work from the Guichard and Hamel labs for centriole maturation. My expertise is related to the fields of centriole and cilia biology.

3. How much time do you estimate the authors will need to complete the suggested revisions:

Estimated time to Complete Revisions (Required)

(Decision Recommendation)

Cannot tell / Not applicable

4. Review Commons values the work of reviewers and encourages them to get credit for their work. Select 'Yes' below to register your reviewing activity at Web of Science Reviewer Recognition Service (formerly Publons); note that the content of your review will not be visible on Web of Science.

Yes

Review #3

1. Evidence, reproducibility and clarity:

Evidence, reproducibility and clarity (Required)

****Summary:**** The manuscript by Ito et al. aims to shed light on the molecular events that form and remodel the engagement link between the existing mother centriole and newly-formed daughter centriole through the cell cycle. Using expansion microscopy, drug treatments, RNAi and CRISPR in HeLa cells, the authors propose that there are four stages of engagement, each regulated by distinct protein complexes, and that inhibition of any stage blocks centriole disengagement at the end of mitosis.

****Major comments:**** Many of the experiments are well-done and interesting. However, additional experiments or analyses are required to support the following major conclusions:

1. A major conclusion of this manuscript is that the engagement linker is remodeled over several stages. This conclusion relies upon drug treatment (HU or HU+Cdk2i), which the authors tell us correspond to 2 distinct stages in early S-phase shortly after procentriole formation. To demonstrate that these treatment conditions accurately represent centriolar events occurring in early S-phase, the authors should:

- a. test whether similar centriole orientations and widths (Fig 1c, d) are observed in untreated early S-phase cells from an asynchronous population
- b. indicate how they determined whether cells in an asynchronous population were in late S-phase or G2 (Fig 1c, 1d, 4d, 4h, 4j)
- c. demonstrate that their inhibitor treatments are arresting cells in the expected stage

2. The authors state that the cartwheel becomes dispensable for maintaining the engagement from Stage 2 to Stage 3 (top of page 12). What is the evidence for this statement? The text points to Fig 2j, but that is a summary cartoon figure.

3. The authors conclude that centriole blooming affects the transition from stage 2 to stage 3 by analyzing cells depleted of Cep295.

- a. Can the authors distinguish between a specific role for Cep295 itself versus a role for centriole blooming? If not, the authors should write that their experiments may also indicate that Cep295 itself plays a role.
- b. What is shown in the bottom panel of Fig 3e? Is this meant to demonstrate a lack of centriole blooming after Cep295 depletion? The dotted yellow line seems to correspond to daughter centriole length, not width.
- c. Do cells depleted of Cep295 fail to expand the distance between the cartwheel and the microtubule wall during blooming? (Fig 3i, j)

4. The authors state that the daughter centriole's proximal end is lost during centriole distancing. This is based on the observation that in G2 phase, alpha-tubulin staining extends more proximally than acetylated tubulin staining, but in mitosis, the gap is lost. The authors propose that the entire proximal end of the centriole is lost in the transition into mitosis.

- a. The authors do not take into account an alternative possibility: that the daughter centriole physically moves further away from the mother centriole and the previously unacetylated microtubules become acetylated. Can the authors rule out this model? If not, the authors should

include this possibility in the text.

- b.If high dose nocodazole truly stabilizes microtubule minus-ends, treatment should result in the daughter centriole being positioned more closely to the mother centriole in mitosis because the minus end is stabilized. The author should test this possibility.

- c.In Fig S5d, the authors use STLC treatment as a control to show loss of the cartwheel. However, STLC arrests cells in prometaphase, while SASS6 is lost from centrioles after metaphase (Strnad et al 2007). STLC treatment is unlikely to be a valid control for this experiment and the authors should replace it.

- d.Does Fig 5e show the same centriole stained for both alpha-tubulin and acetylated tubulin? If not, the authors should redo the analyses of Fig 5e and 5f with the exact same centriole is stained for both alpha-tubulin and acetylated tubulin.

****Minor comments:****

1. In several places in the text, daughter centrioles in early S phase are defined as "thin". The authors should include a numerical value for a "thin" versus "thick" centriole.

2. Significance:

Significance (Required)

Overall, the manuscript is interesting, and the overall conclusions are novel and interesting. The work is helpful for further understanding the mechanisms underlying centriole engagement and disengagement, and the idea that the engagement linker changes from S-phase through mitosis is a valuable conceptual advance to the field. The work extends the authors' previous manuscript on Cep57 and Cep57L1 and add additional mechanistic advances to understanding the overall processes of centriole engagement and disengagement. The major limitations are as stated above in the major comments. The audience will be cell biologists, particularly those interested in centrioles and centrosomes.

3. How much time do you estimate the authors will need to complete the suggested revisions:

Estimated time to Complete Revisions (Required)

(Decision Recommendation)

Between 1 and 3 months

4. Review Commons values the work of reviewers and encourages them to get credit for their work. Select 'Yes' below to register your reviewing activity at Web of Science Reviewer Recognition Service (formerly Publons); note that the content of your review will not be visible on Web of Science.

Yes

Full Revision

Manuscript number: RC-2024-02540

Corresponding author(s): Daiju Kitagawa, Shoji Hata

[Please use this template only if the submitted manuscript should be considered by the affiliate journal as a full revision in response to the points raised by the reviewers.]

*If you wish to submit a preliminary revision with a revision plan, please use our "Revision Plan" template. **It is important to use the appropriate template to clearly inform the editors of your intentions.**]*

1. General Statements [optional]

We thank all the reviewers of our original manuscript for their critical reading and for their useful and constructive comments (typed in black). In response to all the reviewer comments, we have made significant revisions to our manuscript with new data (our responses are typed in blue) as we detail below. We now believe it meets the high-quality standards of EMBO Journal.

This section is mandatory. Please insert a point-by-point reply describing the revisions that were already carried out and included in the transferred manuscript.

Reviewer #1

Evidence, reproducibility and clarity

The article by Ito and colleagues focuses on the mechanisms that govern centriole disengagement. This process is crucial for the function of the centrosome. In G1, the centrosome consists of two mature disengaged centrioles, each of which will initiate a procentriole at the beginning of S-phase. Throughout the S-phase, G2, and up to mitosis, the

Full Revision

procentriole will elongate until it forms a new daughter centriole, which will disengage during mitosis. If this process is altered, the function of the centrosome is compromised and will lead to an abnormal number of centrioles, centrosomes, and mitotic errors.

The mechanisms and molecular players of this engagement are not well known. The same group and others had previously shown that Cep57, Cep57L1, and PCNT were involved in this process. Here, the authors propose a comprehensive study on the disengagement processes and identify several distinct stages during centriole growth: engagement associated with the cartwheel, the torus, and the PCM. The authors also identify factors that regulate the orthogonality of the procentriole, a characteristic that had not been deeply studied before. The description of these stages, coupled with the cell cycles and structural changes of the centriole, provides a better understanding of how this crucial process works.

This work is of high importance for the field. The authors have done meticulous work with a large number of diverse methods, including very well-executed cellular biology approaches coupled with super-resolution microscopy (STED, U-ExM). However, I find that the article is somewhat complex, and some interpretations/conclusions do not always seem correct, or some are overstated (My comments are not ranked in order of importance):

We thank the reviewer for their careful reading of our manuscript and for the positive comments.

- In the introduction, the authors say that there are two centrosomes in G1. Shouldn't there be 1 centrosome with two centrioles instead?

We apologize for any confusion. As described in Tsou et al. (Dev Cell, 2009), centriole disengagement occurs from late mitosis to early G1, resulting in two centrosomes, each containing a single centriole. Therefore, we believe our current description is accurate.

Full Revision

- I am confused between mother centriole, daughter centriole, and procentriole. I think it would be easier to understand if the term procentriole is used until the end of mitosis, giving rise to a daughter centriole in G1, disengaged from the mother. In fact, the disengagement should be the transition from procentriole to daughter centriole.

We appreciate the reviewer's suggestion. While we acknowledge that using 'procentriole' until mitosis might provide clarity, the convention exemplified by studies like Fu et al. (Nat Cell Biol., 2016) designates the newly formed centriole as the 'daughter centriole' throughout its engagement with the mother centriole. The term 'procentriole' may refer to its early stages of formation. Upon disengagement, this daughter centriole is then referred to as the 'new mother centriole.' Our terminology adheres to these conventions.

- The authors link the early steps of procentriole formation and disengagement with the bloom, a recently described structural change of the centriole. In the cited article, the bloom is initiated from the formation of centriole microtubules, whereas here the authors talk about the bloom as a later stage, end of S, which is between stage 2 and stage 3 of the disengagement process described here. It seems to me that the authors should review their analysis and model considering that the bloom is initiated at the beginning of procentriole formation. The bloom step likely appears during stage 1 during orthogonal adjustment.

We thank the reviewer for the insightful comments regarding the timing of the centriole bloom. To address this point, we have conducted new measurements of the daughter centriole's width during cycling S phase (new data presented in Figure 1). The daughter centriole's width measured approximately 141 ± 25 nm in early S phase, which corresponds to the initial stages

Full Revision

of procentriole formation and aligns with the initiation of the bloom as described in the cited work (around 140 nm). This width then increased progressively to 167 ± 26 nm in late S phase and reached 190 ± 18 nm by mitosis, matching the width of the mother centriole. When we arrested cells in early S phase using hydroxyurea (HU, corresponding to stage 2), daughter centriole remained thin (135 ± 13 nm). These findings suggest that centriole blooming does not begin in stage 2 but occurs after stage 2 and during stage 3.

- The authors correlate centriole diameter and the cell cycle (for example in figure 1d). How do the authors know they are in late S and not in G2? If the cells are synchronized, I did not find the information in the materials and methods section.

To determine the specific cell cycle stages, we immunostained the cells with PCNA alongside CP110 and acetylated-tubulin. The distinct staining patterns of PCNA allowed us to differentiate between late S phase and G2 phase. Specifically, dot-like PCNA foci are characteristic of late S phase, while such foci are absent in G2 phase. We will include this information in the figure legends in the revised manuscript (lane 475, 477, 481, and 485 in page 22; lane 513 in page 24; lane 532, 536, and 540 in page 26; lane 559 in page 28).

To relate the cell cycle to procentriole growth, it would be good to correlate diameter with the length of centrioles for a better idea of the procentriole growth stage. This type of analysis could also show that centrioles with HU or HU+Cdk2i treatment have a length that corresponds not to an S phase but rather to a G2 (based on the figures. In figure 3i, the procentriole's length appears correct but in figure 1b procentriole are very long, like in G2).

Full Revision

We appreciate this constructive comment. Following the recommendation, we measured the lengths of centrioles in cells treated with HU or HU+Cdk2i, as well as in cycling cells (new data presented in Figure S1). Our analysis revealed that a subset of centrioles in the treated cells exhibited elongated lengths, comparable to those observed in normal mitosis. Interestingly, despite this elongation, these centrioles remained thinner than those in mitosis, suggesting that they had not yet undergone the blooming process. These findings suggest that the blooming is closely linked to specific stages of cell cycle progression rather than to the extent of procentriole elongation.

- In the same idea of relating the 4 described stages with the cell cycle, stage 2 is probably in late S phase and not early. Therefore, cell synchronization should be done (by mitotic shake-off because treatments with inhibitors may affect centriole length). This analysis would help refine the model and could change some conclusions.

We appreciate the reviewer's suggestion. Stage 2 engagement is defined by the presence of a daughter centriole before blooming and its molecular dependency, which is not solely on Cep57/Cep57L1. Given that mitotic shake-off necessitates cell-cycle arrest, potentially impacting centriole engagement (Karki et al., 2017), we assessed centriole engagement alongside the cell-cycle marker PCNA in untreated cells. In our analysis, we observed thin daughter centrioles in cycling early S phase (new data in Figure 1), similar to those in HU-treated cells, while centrioles in late S phase were thicker. Depletion of Cep57/Cep57L1 did not cause centriole disengagement in cycling early S phase but did in late S phase (Figure s3f, g). Based on these observations, we conclude that stage 2 engagement corresponds to early S phase, not late S phase.

Full Revision

- Regarding the experiment in figure 3e with STED, the authors measured the width using Ac tubulin antibody. Acetylated tubulin staining evolves during growth so if the centrioles are not at the same growth stage, the values can't be comparable. Here procentriole seems to be at different stages. An analysis with tubulin staining would be more accurate. Moreover, I am not sure I understand, is it the width or length that is measured? The line indicates the length.

We appreciate the reviewer's insightful comments regarding Figure 3e and the use of acetylated tubulin staining. We apologize for any confusion caused by the representation in Figure 3e; the line indicates the width, not the length. To clarify this further, we have added a schematic diagram to the figure 3f in the revised manuscript. We believe that measuring the width using this antibody remains appropriate and is not significantly influenced by the centriole's growth stage. Moreover, in Figure 3e, we measured only the daughter centriole width during late S phase, using PCNA staining to ensure comparability.

- Regarding acetylation not covering the proximal part of the procentriole, this result is interesting and suggests a proximal disassembly of the microtubule triplets. Nevertheless, this is staining with two antibodies that do not detect the same epitopes, so the difference could also come from the staining and accessibility. It can also be that tubulin acetylation is gradual and covers the proximal later in the cell cycle. If the tubulin is trimmed on the proximal side, one would imagine that other proximal markers would disappear in G2. For example, is the gamma-TuRC capping the A-microtubule removed? Moreover, the change in angle observed with nocodazole may also be a distancing effect due to the relaxation of the PCM.

Full Revision

We thank the reviewer for their insightful comments regarding the acetylation pattern and proximal disassembly of the daughter centriole. We acknowledge the concern that the observed lack of acetylation at the proximal part of the daughter centriole might be influenced by staining differences or accessibility issues. However, even when using different antibodies from those in the original manuscript, the differences in acetylation patterns persisted, reducing the likelihood that these variations are due to epitope accessibility issues (new data in Figures 5e, f). To further confirm the proximal disassembly of daughter centrioles, we performed additional experiments and have included new data in the revised manuscript.

In response to the question about the gamma-tubulin ring complex (gamma-TuRC) capping the A-microtubule, we found that gamma-TuRC at the proximal end of daughter centriole is indeed removed during mitosis (new data presented in Figure s7a, b). We also observed that the length of the cartwheel structure inside the daughter centriole shrinks at mitotic entry (new data in Figure 5h, i). Importantly, the removals of non-acetylated tubulins, gamma-TuRC, and the cartwheels at the daughter's proximal end were prevented by treatment with a PLK1 inhibitor or high concentrations of nocodazole (new data in Figure 5 e, g, h, i; s7a, b, e, f). These results suggest that the proximal end of the daughter centriole is lost at mitotic entry in a PLK1 and tubulin depolymerization-dependent manner, reinforcing our hypothesis of the proximal disassembly.

Regarding the possibility that the change in angle observed with nocodazole treatment might be due to relaxation of the pericentriolar material (PCM), we examined the PCM structure during mitosis with and without nocodazole. Our new STED data indicate that there were no significant changes in PCM structure during mitosis with nocodazole treatment compared to the control (new data in Figure s7h).

Full Revision

- The authors report that CPAP depletion suppresses precocious centriole disengagement in mitosis under depletion of Cep57 or PCNT. CPAP has also been shown to be important for the formation of microtubule triplets and CPAP depletion leads to incomplete centrioles that fragment (DOI: 10.1083/jcb.202108018). Therefore, I'm not sure to understand the link the authors propose with the loss of the proximal region.

We thank the reviewer for highlighting this important point regarding the role of CPAP. We acknowledge that CPAP is crucial for the formation of microtubule triplets (Vásquez-Limeta et al., 2021). Based on the reviewer's feedback, we developed a hypothesis concerning CPAP's role in centriole blooming, similar to the function of Cep295 (Figure 2). Interestingly, we observed that in CPAP-depleted cells, the centrioles remained narrower than in control cells during late S phase, similar to what we observed with Cep295 depletion (new data in Figure s4a, b). Furthermore, inhibition of CPAP reduced the precocious centriole disengagement phenotype observed in Cep57/Cep57L1-depleted cells (new data in Figure s4c, d). This suggests that CPAP depletion prevents transition of the centriole engagement from stage 2 (cartwheel-cep57-cep57l1 dependent) to stage 3 (cep57-cep57l1 dependent), rather than stage 3 to stage 4 (cep57-dependent), thereby preventing the disengagement phenotype in Cep57/Cep57L1-depleted cells. We have updated the manuscript accordingly and included the new experimental results to support this revised hypothesis (lane 230, page 10).

- The result with Cep63 is very interesting. In the images, it seems that the distance between the wall of the mother centriole and the procentriole is also reduced. Have you quantified this? If the distance is smaller, does this suggest that the procentriole is in phase 1? Does the angle remain obtuse throughout growth?

Full Revision

Although we quantified the distance between the wall of the mother centriole and the daughter centriole, no significant difference was observed between HeLa-Cas9 and Cep63KO cells. To determine if the centriole angle remains obtuse throughout growth, we examined cells in HU-arrested early S phase as well as late S phase and found the angle consistently obtuse (new data is added in Fig. 4h, i). These findings suggest that stable orthogonal engagement does not form in Cep63KO cells. We have updated the manuscript accordingly (lane 301, page 13).

Significance

The article by Ito et al. is very interesting and focuses on the disengagement mechanisms of human centriole. The authors have dissected the process and revealed new, previously unknown steps that follow one another throughout the cell cycle. They have also linked centriole engagement with its orthogonality, a feature of the centriole that is still poorly understood. This article is therefore important for the field. Nevertheless, some analyses do not appear to be fully synchronized with the cell cycle, which may lead to confusion and possibly affect conclusions about the timing of the various steps described.

The major issue concerns the description of timings, which seems to differ from previous observations. An analysis in line with the cell stage would be important here.

We appreciate insightful comments. As mentioned above, we have clarified cell cycle determination using specific cell-cycle markers and included additional analyses of specific cell cycle stages to better outline the timing of each step of engagement configurations in the revised manuscript. We hope these modifications address the reviewer's concerns.

Full Revision

Reviewer #2

Evidence, reproducibility and clarity

Summary

This study explores the events of centriole engagement and disengagement. Centriole engagement dynamics are important because engagement ensures that centriole overduplication does not occur, but cells must also be prepared for disengagement during mitosis to enable centriole duplication during G1/S of the following cell cycle. This study characterizes four stages for centriole engagement dynamics: 1) Stage 1: at G1/S boundary with flexible connection of the daughter centriole to the mother centriole wall becoming orthogonal, 2) Stage 2: promotes blooming of centriole width that, based on prior studies, suggesting centrioles of singlet MTs gaining triplet MTs, 3) Stage 3: further distancing by loss of the proximal centriole during mitosis. Finally, disengagement occurs during separase cleavage in mitosis. The cartwheel, torus, and PCM and associated molecules are all required for unique and overlapping phases of the engagement. While some of these stages and events are reported in prior work, this study provides a comprehensive view of centriole engagement. Overall, the study is thorough and provides important insights for how centriole engagement/disengagement occurs. In places, I struggled with the complexity of the manuscript as currently written, and provide details for how I believe the manuscript can be improved below.

We thank this reviewer for the thorough review and positive feedback. We have modified complex sections as suggested and made revisions to enhance readability and precision throughout the manuscript. We hope these improvements address the concerns.

Major

Full Revision

Generally, the manuscript is written for the expert and could be improved by simplifying the text and findings and clearly explaining the various perturbations and treatments provided. It was easy to get bogged down in the details of this manuscript. Clearer/tighter writing would improve the impact of the paper. Reader is asked to wade through a lot of jargon and detail of the experiments to follow. There is a lot of "we did..", I wonder if this can be told as a more simple story to really get things going. Perhaps having an outside reader help to shorten the text to more concise statements would address this issue.

We appreciate the reviewer's feedback regarding the clarity and writing style of the manuscript. In response, we have revised the text to reduce jargon and simplify explanations, focusing on presenting the key findings more clearly and concisely.

If CycE/CDK2 is promoting stabilization of the new daughter centriole into an orthogonal position, what is promoting new centriole assembly. Is Plk4 only required for assembly in the canonical cell cycle, while CycE/CDK2 is not required for new centriole assembly? This delineation of roles for new centrioles forming independent of the CDK2 and cyclin E and/or A activity seems important to describe.

We thank the reviewer for this constructive suggestion. Our findings demonstrate that while CycE/CDK2 promotes the stabilization of the new daughter centriole into an orthogonal position and CycA/CDK2 initiates centriole blooming, the new centriole assembly is not affected by these complexes, but is primarily driven by Plk4. We have described the role of CDK2 complexes in the result section of the revised manuscript (lane 170, page 8).

Full Revision

Perhaps most surprising is the loss of the proximal end of the centriole. This is a non-acetylated tubulin domain. Do centrioles get shorter?

To address this point, we measured the length of the daughter centriole before and after distancing (G2 vs prometaphase). Interestingly, daughter centrioles in mitosis were longer than those in the G2 phase (G2: 136 ± 12.1 nm, prometaphase: 282 ± 35.2 nm). This observation aligns with findings from Kong et al. (2020), who demonstrated that the daughter centriole can elongate during mitosis. We assume that while the proximal part of the centriole is lost during mitosis (new data in Figure 5, s6, s7), the elongation of the distal parts compensates for this loss, resulting in an overall increase in centriole length during mitosis.

I think this needs to be flushed out more by costaining MTs generally and then the Ac Tub specifically. The proposal that the proximal region of the centriole and CW is disassembled is interesting, however I don't think that this is well established from these data. First, I do not see support for this from prior studies, like Laporte et al 2024, who co-localized Actubulin and tubulin. From these studies, it appears that Actubulin is at the proximal region early in assembly and is maintained there. Can the authors study both markers in the context of their perturbations to establish this?

We appreciate this suggestion. In response, we performed co-immunostaining of alpha-tubulin and acetylated tubulin using U-ExM. Using the antibodies from Laporte et al. (2024), both alpha-tubulin and acetylated tubulin showed the similar localization pattern at the proximal region as reported. However, when we used the acetylated tubulin antibody from our original Figure 5 (Figure s6e, f in the revised manuscript), we observed significant differences between alpha-

Full Revision

tubulin and acetylated tubulin at the proximal end of the daughter centriole, indicating the existence of non-acetylated tubulin (new data in Figure 5e-g). This difference was not seen in mitotic daughter centrioles or mother centrioles. Moreover, treatment with PLK1 inhibitor or nocodazole prevented the loss of the non-acetylated tubulin region (new data in Figure 5e-g). Considering these results along with the loss of γ -TuRC and the shrinkage of the cartwheel at mitotic entry (as mentioned below), we propose that the proximal region of the daughter centriole is indeed lost as cells enter mitosis.

Are the gamma-tubulin capped minus ends of the A-MTs lost?

Using U-ExM, we observed gamma-tubulin caps at the ends of daughter centrioles in interphase (new data in Figure s7a, b). However, during mitosis, the increased gamma-tubulin signal in the pericentriolar material made it challenging to assess capping at the minus ends. To address this, we induced mitotic exit by treating cells with a CDK1 inhibitor and examined the daughter (or new mother) centrioles. We found that the gamma-tubulin cap was lost during mitosis, and this loss was prevented by PLK1 inhibitor treatment (new data in Figure s7a, b). This suggests that the gamma-tubulin caps at the minus ends are indeed lost during mitosis in PLK1-dependent manner.

What is the predicted length for the CW and does that correlate with the 30nm of MTs that is proposed to be lost?

We appreciate this point. We measured the length of the cartwheel structure inside the daughter centriole and observed a shrinkage of approximately 20 nm at mitotic entry (new data in Figure

Full Revision

5h, i). Importantly, this shrinkage was prevented by treatment with a PLK1 inhibitor or high concentrations of nocodazole. Although the shrinkage is a little bit shorter than the non-acetylated microtubule loss at mitotic entry (24.8 nm, new data in Figure 5g), this observation also supported the proximal loss of the daughter centrioles.

-

Do the authors observe the centriole to shorten at this step of separation?

As mentioned above, the centriole was not shortened in mitosis.

Beyond the above comments, I think a clear Discussion of what might promote this and how it could be controlled is necessary.

Based on the new data (Figures 5, s6, s7), we added some discussion in the result section of the revised manuscript as follows (lane 351, page 16):

It is possible that PLK1 might cause the removal of the γ -tubulin cap, leading to microtubule depolymerization at the daughter's proximal end.

Does the high nocodazole treatment affect cell cycle progression, and could that produce the described result of no separation/distanced centrioles? Could high drug treatment arrest the cells earlier and this blocks "distancing"? Please address.

Full Revision

We appreciate the reviewer's insightful comments. As the reviewer correctly noted, Nocodazole treatment arrests cells in prometaphase. To address this concern, we conducted additional experiments using STLC, which also arrests cells at the prometaphase stage. Under STLC conditions, we observed that distancing still occurred, and the angle between centrioles was altered (new data in Figure 5e, f, s7c, d). Therefore, we believe that the suppression of centriole distancing observed with high-dose Nocodazole is not due to cell cycle arrest but rather a specific effect of the drug on centriole behavior.

It was not clear to me that the Cdk2i+HU resulted in an earlier arrest. Was this timing of the cell cycle confirmed? Are there markers that the authors could use beyond just the statement that orthogonal positioning is now changed? Or can you show this by FACS analysis of DNA replication? Or centrin maturation?

We thank the reviewer for the insightful comment. In response, we performed immunostaining and FACS analysis. The immunostaining results showed that the signal for p-Rb (Ser807/811), which is phosphorylated first by Cdk4 and then further by Cdk2 during the G1/S transition, was reduced under Cdk2i + HU treatment compared to HU alone (new data in Fig.s1a, b). FACS analysis further confirmed that cells treated with Cdk2i + HU were arrested at an earlier stage than those treated with HU alone, based on their DNA content (new data in Fig. s1c). These results together support our conclusion that Cdk2i + HU causes an earlier arrest in the cell cycle.

Methods for measurements: The methods for measuring and calibrating to the ExM samples needs to be explained in the Materials and Methods section. For one example, how was

Full Revision

centriole width defined? I assume that this was based on a full width half maximum of a certain criterion. This should be defined.

The centriole width was defined as the position where the signal intensity dropped to half of the maximum at the outer edge of the centriole cylinder. In the revised manuscript, we have included this detail in the material and method section (lane 840, page 41).

Figure 2C,D - 30% of the cells exhibited precocious disengagement. What happened to the remaining 70%?

In the remaining 70% of cells, while the cartwheel structure was lost, two pairs of centrioles that maintained engagement were observed as in control cells. A reduction in centriole numbers was rarely observed in these cells (as shown in Fig. 2b).

Figure 2F - Why did the authors focus on quantifying 80-100deg and not report the SD and coefficient of variation? It seems like this would be more accurate and a stronger description of the variance?

Following the comment, we updated our analysis in the revised Figure 2g by using a box plot, and we have also included the standard deviation and coefficient of variation.

Despite identifying cell cycle (cycE/CDK2) and molecular factors (Cep63) responsible, it would be interesting if the Discussion focused more on how the unique events of Stage 1 to Stage 2 occurs. What is a mechanistic model for this?

Full Revision

We appreciate the reviewer's insightful comments, and we have revised the Discussion to address the points raised by the reviewer. The Discussion now includes the following (lane 442.

Page 20):

In light of the role of the CDK2-cyclin E complex in orthogonal engagement formation in early S phase, we propose that this kinase complex may specifically target the torus proteins to initiate engagement.

Similarly, why would blooming stabilize the links? I think this should at least be a discussion point.

Although this reviewer mentions that blooming stabilizes the engagement, this event actually renders the cartwheel structure dispensable for centriole engagement possibly by increasing the distance between daughter centriole wall and cartwheels, as detailed in Figure 3k.

I found this same theme for the codependences between Stage 3 factors as well. It was not clear why these factors would be codependent. I think this can also be addressed in the discussion.

While this reviewer mentions the codependence of Stage 3 factors, the torus and the PCM, they redundantly maintain centriole engagement in this Stage 3. The redundancy between torus proteins and PCM factors is addressed in Figures 5l–m. This redundancy may result from the spatial relationships among the torus, PCM, and daughter centriole. Both the torus and PCM

Full Revision

can interact with the daughter centriole during interphase, but in mitosis, only the PCM can interact and maintain the engagement.

Minor

Define the torus and PCM for general readers.

We have added explanations for the torus and PCM in the introduction as follows:

In the G1 phase, cells contain two centrosomes, each with a mature centriole (mother centriole) surrounded by a protein matrix that includes the torus and the pericentriolar material (PCM).

The torus, located just outside the centriole wall, supports centriole duplication, while the PCM surrounds the torus and organizes microtubules, facilitating MTOC activity⁵

To be clearer, I would suggest that "after cell division" is stated at "during G1" or whatever the authors intend. I found this statement could be more precise. Same for "end of mitosis."

We appreciate the reviewer's suggestion. We have rephrased 'after cell division' and 'end of mitosis' to 'G1 phase' for clarity and precision.

There are some minor grammatical mistakes that should be addressed.

We have carefully reviewed and revised the manuscript to correct minor grammatical mistakes throughout.

Full Revision

There are a number of places where the references should be expanded to provide context for these results relative to the literature in the field. An example is the CW engagement role. Guichard et al 2010 EMBO have structural data showing such an attachment. We appreciate the reviewer's suggestion. We have expanded the references in the manuscript, including Guichard et al 2010 EMBO article.

How does centrinone treatment remove the CW? This was not well explained. In the revised manuscript, we included the findings from Kim et al. (2016, Cell Reports), which demonstrated that persistent PLK4 activity is essential for maintaining the integrity of the cartwheel structure, as shown through PLK4 inhibition experiments.

I was unclear as to why in some places "Cdk2" was used and others "CDK2" was used. Please keep consistent.

We apologize for the inconsistency in the usage of "Cdk2" and "CDK2." We have revised the manuscript to use "CDK2" consistently throughout the text to avoid any confusion.

I was not clear how Figure 2E showing "CDK2 is activated by forming complexes....". Please clarify.

In the revised manuscript, we clarified this point as follows:

CDK2 is activated by forming complexes with either cyclin E or cyclin A²⁶. Immunofluorescence analysis revealed that cyclin E expression increases during late G1/early S phase (Fig. 2e),

Full Revision

while cyclin A expression rises during late S phase (Fig. 3a). This temporal expression pattern implies that CDK2-cyclin E activity may be responsible for initiating the transition from Stage 1 to Stage 2.

Significance

Overall, this is an interesting paper that identifies important findings associated with the field of centriole engagement and disengagement dynamics. This will be important for the centriole field but may be less interesting to a general audience. The work appears to be associated with prior studies from the Loncerak lab on Plk1 control of centriole separation during mitosis, the Rhee lab, and others. Some of this work relates to recent work from the Guichard and Hamel labs for centriole maturation. My expertise is related to the fields of centriole and cilia biology

We thank the reviewer for their positive feedback.

Full Revision

Reviewer #3 (Evidence, reproducibility and clarity (Required)):

Summary: The manuscript by Ito et al. aims to shed light on the molecular events that form and remodel the engagement link between the existing mother centriole and newly-formed daughter centriole through the cell cycle. Using expansion microscopy, drug treatments, RNAi and CRISPR in HeLa cells, the authors propose that there are four stages of engagement, each regulated by distinct protein complexes, and that inhibition of any stage blocks centriole disengagement at the end of mitosis.

We thank the reviewer for the thoughtful comments.

Major comments: Many of the experiments are well-done and interesting. However, additional experiments or analyses are required to support the following major conclusions:

1) A major conclusion of this manuscript is that the engagement linker is remodeled over several stages. This conclusion relies upon drug treatment (HU or HU+Cdk2i), which the authors tell us correspond to 2 distinct stages in early S-phase shortly after procentriole formation. To demonstrate that these treatment conditions accurately represent centriolar events occurring in early S-phase, the authors should:

a. test whether similar centriole orientations and widths (Fig 1c, d) are observed in untreated early S-phase cells from an asynchronous population

We thank this reviewer for the constructive comment. To address this concern, the orientation and the width of daughter centrioles were analyzed in untreated S-phase cells identified by a PCNA staining. The results showed that untreated cells showed an unstable angle in early S

Full Revision

phase, but not in late S phase (new data in Figure 1c). Additionally, the daughter centriole width was thinner in untreated early S phase than in late S phase (new data in Figure 1d).

b.indicate how they determined whether cells in an asynchronous population were in late S-phase or G2 (Fig 1c, 1d, 4d, 4h, 4j)

PCNA staining was used to identify late S-phase cells, and CENP-F staining was employed to identify G2 phase cells. This information has been added to the figure legends in the revised manuscript (lane 475, 477, 481, and 485 in page 22; lane 513 in page 24; lane 532, 536, and 540 in page 26; lane 559 in page 28).

c.demonstrate that their inhibitor treatments are arresting cells in the expected stage

To confirm that the inhibitor treatments arrest cells at the expected stages, FACS analysis was performed. The results indicated that HU treatment arrests cells in early S-phase, while Cdk2i + HU arrests cells even earlier, near the G1/S transition, based on DNA content (new data in Figure s1c). Immunostaining further supported these findings: phosphorylation of pRb-407/408, which occurs during the G1/S transition, was reduced under Cdk2i + HU compared to HU alone (new data in Figure s1a, b). These results demonstrate that the treatments effectively arrest the cell cycle at the intended stages.

2)The authors state that the cartwheel becomes dispensable for maintaining the engagement from Stage 2 to Stage 3 (top of page 12). What is the evidence for this statement? The text points to Fig 2j, but that is a summary cartoon figure.

Full Revision

The evidence comes from observations that in Stage 2, disengagement occurs only when Cep57, Cep57L1, and the cartwheel are all depleted (Fig. 2c, d). In contrast, in Stage 3, disengagement happens with the depletion of Cep57 and Cep57L1 alone (Fig. 1e, f), indicating that the cartwheel is no longer required for maintaining engagement by Stage 3. To clarify this point, the description in the manuscript has been revised as follows (lane 214, page 10):

During late S phase, the configuration and mechanism of centriole engagement change: daughter centrioles undergo blooming, and the engagement mechanism transitions from a cartwheel-Cep57-Cep57L1-dependent state (Stage 2) to a Cep57-Cep57L1-dependent state (Stage 3) (Fig. 2j). In other words, the cartwheel becomes dispensable for maintaining engagement.

3)The authors conclude that centriole blooming affects the transition from stage 2 to stage 3 by analyzing cells depleted of Cep295.

a.Can the authors distinguish between a specific role for Cep295 itself versus a role for centriole blooming? If not, the authors should write that their experiments may also indicate that Cep295 itself plays a role.

To address whether the observed effects are due to centriole blooming or a specific role of Cep295, similar assays were performed using CPAP, a protein known to stabilize nascent centrioles. Like Cep295 depletion, siCPAP treatment inhibited blooming and prevented the centriole disengagement caused by Cep57 and Cep57L1 depletion (new data in Figure s4). These results suggest that blooming rather than Cep295 itself is the key factor affecting the transition from Stage 2 to Stage 3.

Full Revision

b. What is shown in the bottom panel of Fig 3e? Is this meant to demonstrate a lack of centriole blooming after Cep295 depletion? The dotted yellow line seems to correspond to daughter centriole length, not width.

We apologize for the confusion. In the bottom panel of Fig. 3e, the dotted yellow line represents the width of the daughter centriole, demonstrating the lack of centriole blooming after Cep295 depletion. To clarify this, a schematic diagram has been added to the figure 3f.

c. Do cells depleted of Cep295 fail to expand the distance between the cartwheel and the microtubule wall during blooming? (Fig 3i, j)

Using U-ExM, we observed that depletion of Cep295 resulted in a failure to increase the distance between the cartwheel and the microtubule wall (new data is in Figure 3i, j, k).

4) The authors state that the daughter centriole's proximal end is lost during centriole distancing. This is based on the observation that in G2 phase, alpha-tubulin staining extends more proximally than acetylated tubulin staining, but in mitosis, the gap is lost. The authors propose that the entire proximal end of the centriole is lost in the transition into mitosis.

a. The authors do not take into account an alternative possibility: that the daughter centriole physically moves further away from the mother centriole and the previously unacetylated microtubules become acetylated. Can the authors rule out this model? If not, the authors should include this possibility in the text.

Full Revision

We acknowledge the alternative possibility raised by the reviewer—that the daughter centriole may move further away from the mother centriole and that previously non-acetylated microtubules become acetylated. However, our new results demonstrate that the structures at the proximal ends, including the cartwheel, γ -TuRC capping A-tubule and non-acetylated microtubules, were lost in mitosis (new data in Figures 5, s6, s7). Furthermore, these reductions were suppressed by preventing minus-end depolymerization of tubulins with a high-dose nocodazole treatment (new data in Figure 5, s7). We therefore propose that the proximal region of the daughter centriole is indeed depolymerized as cells enter mitosis.

b.If high dose nocodazole truly stabilizes microtubule minus-ends, treatment should result in the daughter centriole being positioned more closely to the mother centriole in mitosis because the minus end is stabilized. The author should test this possibility.

In response to the reviewer's suggestion, we measured the distance between the mother centriole wall and the proximal end of the daughter centriole in mitotic cells treated with high-dose nocodazole or PLK1 inhibitor. Our results show that both treatments indeed result in the daughter centriole being positioned more closely to the mother centriole than control cells (new data in Figure 5e, f).

c.In Fig S5d, the authors use STLC treatment as a control to show loss of the cartwheel. However, STLC arrests cells in prometaphase, while SASS6 is lost from centrioles after metaphase (Strnad et al 2007). STLC treatment is unlikely to be a valid control for this experiment and the authors should replace it.

Full Revision

We appreciate the reviewer's concern. While it is known that the cartwheel is typically lost after metaphase (Strnad et al., 2007), our observations indicate that 3 hours STLC treatment, which arrests cells in prometaphase, also leads to cartwheel loss (Fig. s7f, g). This suggests that prolonged prometaphase arrest can induce premature cartwheel disassembly. Since both nocodazole and PLK1 inhibitors also arrest cells in prometaphase, we believe that STLC treatment remains an appropriate control for assessing cartwheel retention under these conditions. We have clarified this rationale in the revised manuscript (line 165, page 8) as follows:

Given that the maintenance of the cartwheel depends on persistent PLK4 activity²⁵, the PLK4 inhibitor centrinone was added after daughter centriole formation.

d. Does Fig 5e show the same centriole stained for both alpha-tubulin and acetylated tubulin? If not, the authors should redo the analyses of Fig 5e and 5f with the exact same centriole stained for both alpha-tubulin and acetylated tubulin.

The reviewer is correct that the original images did not show the same centriole. In response, we have performed new experiments using U-ExM to obtain images of the same centriole stained for both alpha-tubulin and acetylated tubulin. The updated images and analyses have been included in Figures 5e.

Minor comments:

1) In several places in the text, daughter centrioles in early S phase are defined as "thin". The authors should include a numerical value for a "thin" versus "thick" centriole.

Full Revision

We appreciate the suggestion. In the revised manuscript, we have specified that "thin" daughter centrioles in early S phase have a diameter of less than 160 nm, while "thick" centrioles in later stages have a larger diameter (lane 253, page 11).

Reviewer #3 (Significance (Required)):

Overall, the manuscript is interesting, and the overall conclusions are novel and interesting. The work is helpful for further understanding the mechanisms underlying centriole engagement and disengagement, and the idea that the engagement linker changes from S-phase through mitosis is a valuable conceptual advance to the field. The work extends the authors' previous manuscript on Cep57 and Cep57L1 and add additional mechanistic advances to understanding the overall processes of centriole engagement and disengagement. The major limitations are as stated above in the major comments. The audience will be cell biologists, particularly those interested in centrioles and centrosomes.

We thank this reviewer for their positive feedback and are pleased that this reviewer finds our work novel and valuable to the field of cell biology.

Prof. Daiju Kitagawa
The University of Tokyo
Laboratory of Physiological Chemistry
Graduate School of Pharmaceutical Sciences
Bunkyo, Tokyo 113-0033
Japan

29th Nov 2024

Re: EMBOJ-2024-119339-T
Multimodal mechanisms of human centriole engagement and disengagement

Dear Daiju,

Thank you for submitting your revised Review Commons manuscript for consideration by The EMBO Journal. Given the interest of the subject and the constructive nature of the transferred referee reports, I decided to treat the work as a regular EMBO Journal revision, and returned it directly to the original referees 1 and 3. I am happy to say that both consider the manuscript significantly strengthened and now in principle suited for publication. Referee 3 nevertheless retains a few specific concerns (see below), which I would invite you to address during a final round of minor revision.

At this stage, please also take care of the following editorial issues, mainly to adjust the manuscript format according to EMBO Journal guidelines:

GENERAL:

- Please download and complete our author checklist (link provided below).
- Please provide suggestions for a short 'blurb' text prefacing and summing up the conceptual aspect of the study in two sentences (max. 250 characters), followed by 3-5 one-sentence 'bullet points' with brief factual statements of key results of the paper; they will form the basis of an editor-written 'Synopsis' accompanying the online version of the article. Please also upload a synopsis image, which can be used as a "visual title" for the synopsis section of your paper (maybe based on Figure 6E?). The image should be in PNG or JPG format, and please make sure that it remains in the modest dimensions of (exactly) 550 pixels wide and 300-600 pixels high.

TEXT:

- Please adjust the order of the manuscript sections: Title page with complete author information, Abstract, Keywords, Introduction, Results, Discussion, Methods, Data Availability, Acknowledgements, Disclosure and Competing Interests Statement, References, Main Figure Legends, Tables, Expanded Figure Legends.
- On the abstract page of the manuscript, please include 4-5 general keyword terms to enhance searchability.
- Please note that Materials and Methods need to be described in the main text using our 'Structured Methods' format (for detail, see <https://www.embopress.org/page/journal/14693178/authorguide#structuredmethods>). The in-text "Methods" section should contain method and protocol descriptions (ideally using a step-by-step protocol format to facilitate adoption of the methodologies across labs), while all key reagents, experimental models, software and relevant equipment - including their sources and relevant identifiers - should be listed in a separately uploaded Reagents and Tools Table, a template for which can be downloaded from the above section of our Author Guidelines.
- As we are switching from a free-text author contribution statement towards a more formal statement based on Contributor Role Taxonomy (CRediT) terms, please remove the present Author Contribution section and instead specify each author's contribution(s) directly in the Author Information page of our submission system during upload of the final manuscript. See <https://casrai.org/credit/> for more information.
- Please rename the Competing Interest section into "Disclosure and Competing Interests Statement", in accordance with our updated Guide to Authors (<https://www.embopress.org/competing-interests>)
- Please adjust the format of the reference list and of the in-text citations according to EMBO Journal format (alphabetical order, author name et al + year, first up to 10 authors should be listed, followed by 'et al' ...).
- Please include a dedicated "Data Availability" section at the end of the Material and Methods (suggested wording: "The

[structural coordinates | microarray | mass spectrometry] data from this publication have been deposited to the [name of the database] database [URL] and assigned the identifier [accession | permalink | hashtag]."); should there no data deposition to public repositories linked to the study, this should still be stated as "This study includes no data deposited in external repositories."

- Please make sure to include all relevant funding information not only in the manuscript text, but also in our submission system.

DATA:

- Please refer to our author guide (www.embopress.org/page/journal/14602075/authorguide#expandedview) regarding "supplementary information", and consider re-organizing the current figures and supplemental figures. We are not limited to 6 main figures, and in addition, we can have up to 5 "Expanded View" figures, whose legends would also need to be in the main text, and which would be type-set and directly visible (expandable) with the HTML version of the paper. My suggestion would be to convert 5 of the current supplementary figures into Expanded View figures (naming/in-text callouts: Figure EV1-5), and to promote some other "supplementary" content into additional main figures/panels. Any remaining "supplementary" content should be provided in a single "Appendix" PDF (please refer to the above-referenced section of our author guidelines for detailed information).

- All main and EV figures should be uploaded as individual files with sufficient resolution/quality for production. Please also make sure to call-out all of the individual Figure panel at least once in the text.

- Finally, during routine pre-acceptance checks, our data editors have raised the following queries regarding figures, data, and legends; I would appreciate if you briefly answered to them in the cover letter of your final submission, and made the requested text modifications with changes/additions highlighted via the "Track changes" option, to facilitate our final checking.

1. Please note that the exact p values are not provided in the legends of figures 1H, 2B, D, I; 3C, D, H; 4F, I; 5B, F, G, I, K; 6C.
2. Please indicate the statistical test used for data analysis in the legends of figures 2B, I; 4F, K; 5D, F, G, I, K; 6C.
3. Please note that the box plots need to be defined in terms of minima, maxima, centre, bounds of box and whiskers, and percentile in the legends of figures 1C, F; 2G, 3F, 4B, F, 5F, G, I, M.
4. Please note that the scale bar needs to be defined for figures 5J, 6B
5. Please note that the white arrow heads are not defined in the legend of figures 1G, 3B, 5J, 6B. This needs to be rectified.
6. Please note that the white two-way arrows are not defined in the legend of figure 6A. This needs to be rectified.

Should you need additional guidance/feedback regarding this final adjustments, please do not hesitate to contact us directly. Thank you again for the opportunity to consider this work for The EMBO Journal, and I look forward to receiving your re-revised manuscript.

With kind regards,

Hartmut

9) To facilitate reproducibility and cross-laboratory adoption of methodologies, please structure the Materials & Methods section as outlined in our guide to authors, including a completed Reagents and Tools Table that can be downloaded from our author guidelines as well (<https://www.embopress.org/page/journal/14602075/authorguide#structuredmethods>).

10) Digital image enhancement is acceptable practice, as long as it accurately represents the original data and conforms to community standards. If a figure has been subjected to significant electronic manipulation, this must be clearly noted in the figure legend and/or the 'Materials and Methods' section. The editors reserve the right to request original versions of figures and the original images that were used to assemble the figure. Finally, we generally encourage uploading of numerical as well as gel/blot image source data; for details see: embopress.org/page/journal/14602075/authorguide#sourcedata

At EMBO Press, we ask authors to provide source data for the main manuscript figures. Our source data coordinator will contact you to discuss which figure panels we would need source data for and will also provide you with helpful tips on how to upload and organize the files.

In the interest of ensuring the conceptual advance provided by the work, we recommend submitting a revision within 3 months (27th Feb 2025). Please discuss the revision progress ahead of this time with the editor if you require more time to complete the revisions. Use the link below to submit your revision:

Link Not Available

Referee #1:

I am satisfied with the revisions, which have improved the clarity of the article and strengthened its conclusions. I am pleased to recommend this article for publication.

Referee #3:

I commend the authors for these excellent revisions. For the most part, my comments from the original review have been addressed. There are 3 remaining issues, 1 major and 2 minor, that the authors should clarify before final publication. In all other respects, I believe that the manuscript is much improved and will be of great interest to cell biologists, especially those interested in centrosomes and cilia. Congratulations on an excellent and thought-provoking study.

1. Major: In Fig 5e, I am glad that the authors now show acetylated tubulin and alpha-tubulin for the same centrioles as

suggested. However, the authors should add an additional panel for a mitotic centriole that has not been drug-treated, to demonstrate that the daughter centriole has distanced itself from the mother. Right now, the panels only show interphase, STLC treated, high-dose nocodazole treated, and PLK1i treated, and thus it is important to show that distancing occurs normally in mitosis, even without drug treatment.

2. Minor: In Fig 3e, S4a, it is still difficult for me to understand how the yellow lines mark daughter centriole width. In my version of the figure, there are 2 yellow lines in the control, 3 in the siCCNA2 and 3 in siCep295. Each yellow line goes all the way across the entire box of the inset image and seems unrelated to the width of the centriole. It would be helpful for readers if the line more clearly marked the width of the centriole.

3. Minor: in the author's response to reviewers document, in response to 4c) about STLC treatment, they mention that they clarify the rationale for STLC use in line 165, page 8. However, this sentence refers to centrinone treatment and not STLC, and I did not find additional information in the revised manuscript about the justification for STLC use. The authors should include the information in the manuscript.

Rev_Com_number: RC-2024-02540

New_manu_number: EMBOJ-2024-119339-T

Corr_author: Kitagawa

Title: Multimodal mechanisms of human centriole engagement and disengagement

We thank all the reviewers of our manuscript for their critical reading and for their useful and constructive comments (typed in black). In response to all the reviewer comments, we have made significant revisions to our manuscript with new data (our responses are typed in blue) as we detail below. We now believe it meets the high-quality standards of EMBO Journal.

Referee #1:

I am satisfied with the revisions, which have improved the clarity of the article and strengthened its conclusions. I am pleased to recommend this article for publication.

We thank this reviewer for the positive comment.

Referee #3:

I commend the authors for these excellent revisions. For the most part, my comments from the original review have been addressed. There are 3 remaining issues, 1 major and 2 minor, that the authors should clarify before final publication. In all other respects, I believe that the manuscript is much improved and will be of great interest to cell

biologists, especially those interested in centrosomes and cilia. Congratulations on an excellent and thought-provoking study.

We thank this reviewer for the positive comment and suggestion.

1. Major: In Fig 5e, I am glad that the authors now show acetylated tubulin and alpha-tubulin for the same centrioles as suggested. However, the authors should add an additional panel for a mitotic centriole that has not been drug-treated, to demonstrate that the daughter centriole has distanced itself from the mother. Right now, the panels only show interphase, STLC treated, high-dose nocodazole treated, and PLK1i treated, and thus it is important to show that distancing occurs normally in mitosis, even without drug treatment.

As requested, we have included an additional panel in Fig. 5e, showing a mitotic centriole that has not been subjected to drug treatment. This panel demonstrates that daughter centrioles naturally distance themselves from the mother centrioles during mitosis under normal conditions.

2. Minor: In Fig 3e, S4a, it is still difficult for me to understand how the yellow lines mark daughter centriole width. In my version of the figure, there are 2 yellow lines in the control, 3 in the siCCNA2 and 3 in siCep295. Each yellow line goes all the way across the entire box of the inset image and seems unrelated to the width of the centriole. It would be helpful for readers if the line more clearly marked the width of the centriole.

We thank the reviewer for pointing out the difficulty in interpreting the yellow lines used to indicate daughter centriole width in Fig. 3e and Fig. S4a. To improve clarity, we have replaced the yellow dashed lines with bidirectional arrows that more accurately mark the width of the centriole.

3. Minor: in the author's response to reviewers document, in response to 4c) about STLC treatment, they mention that they clarify the rationale for STLC use in line 165, page 8. However, this sentence refers to centrinone treatment and not STLC, and I did not find additional information in the revised manuscript about the justification for STLC use. The authors should include the information in the manuscript.

We thank the reviewer for pointing out the error in our response regarding the rationale for STLC treatment. We have now included a justification for the use of STLC in the revised manuscript (line 371, page 17) as follows:

We also examined the effect on the cartwheel in prometaphase arrested cells. Although the cartwheel is known to be degraded after metaphase (Strnad et al., 2007), our observations indicate that prolonged prometaphase arrest also leads to cartwheel loss (Fig. EV5C,D; STLC treatment: cartwheel retention, $18.9 \pm 6.9\%$). High-dose nocodazole treatment, however, effectively suppressed this cartwheel loss in prometaphase-arrested cells (Fig. EV5C,D; $48.8 \pm 15.7\%$).

Dear Prof. Kitagawa,

I am pleased to inform you that your manuscript has been accepted for publication in the EMBO Journal.

Yours sincerely,

Rev_Com_number: RC-2024-02540

New_manu_number: EMBOJ-2024-119339R

Corr_author: Kitagawa

Title: Multimodal mechanisms of human centriole engagement and disengagement